# Mysteries of the Deep: Role of Intermediate Representations in Out of Distribution Detection

**Ignacio Meza De la Jara**[1,4]    **Cristian Rodriguez-Opazo**[1]    **Damien Teney**[1,2]
**Damith Ranasinghe**[1]    **Ehsan Abbasnejad**[1,3]
[1]Australian Institute for Machine Learning, University of Adelaide
[2]Idiap Research Institute, Switzerland    [3]Monash University    [4]Naval Group
{firstname.lastname}@adelaide.edu.au

## Abstract

Out-of-distribution (OOD) detection is essential for reliably deploying machine learning models in the wild. Yet, most methods treat large pre-trained models as monolithic encoders and rely solely on their final-layer representations for detection. We challenge this wisdom. We reveal the *intermediate layers* of pre-trained models, shaped by residual connections that subtly transform input projections, *can* encode *surprisingly rich and diverse signals* for detecting distributional shifts. Importantly, to exploit latent representation diversity across layers, we introduce an entropy-based criterion to *automatically* identify layers offering the most complementary information in a training-free setting—*without access to OOD data*. We show that selectively incorporating these intermediate representations can increase the accuracy of OOD detection by up to $10\%$ in far-OOD and over $7\%$ in near-OOD benchmarks compared to state-of-the-art training-free methods across various model architectures and training objectives. Our findings reveal a new avenue for OOD detection research and uncover the impact of various training objectives and model architectures on confidence-based OOD detection methods. 🌐

## 1   Introduction

Out-of-distribution (OOD) detection is essential for reliable machine learning, especially in open-world settings with distribution shifts that risk unsafe predictions [21, 55]. The problem is acute in safety-critical domains such as autonomous driving, medical diagnostics, and cybersecurity [17, 71].

Recent work has leveraged large vision–language models (VLMs) such as CLIP [57] in attempts to address the problem. The methods enable zero-shot OOD detection by aligning image and text embeddings. However, the approaches implicitly treat these deep models as shallow because the information extracted for detection simply focus on the last layer. Indeed, deep neural networks typically rely on final-layer embeddings as compact, semantically rich representations of the input. But, a sole reliance on the last layer overlooks the neural structure through which they are obtained. Therefore, we challenge this widespread wisdom and propose exploring intermediate-layer representations.

Interestingly, early work in convolutional models showed distinct functionality across layers [2] however pretrained vision transformers trained with diverse objectives—supervised, contrastive, or masked modeling—have not been thoroughly examined in this context. Understanding how in- and out-of-distribution semantics are distributed across *depth* may offer means for improving detection robustness and generalization. Our study revisits the problem of zero-shot OOD detection to ask: *Can intermediate representations be systematically leveraged to improve detection performance?*

To investigate, we conduct a comprehensive analysis across seven vision backbones, including CLIP, DINOv2 [54], MAE [23], MoCo-v3 [8], SiGLiP-v2 [61], Perception Encoder [4], and supervised

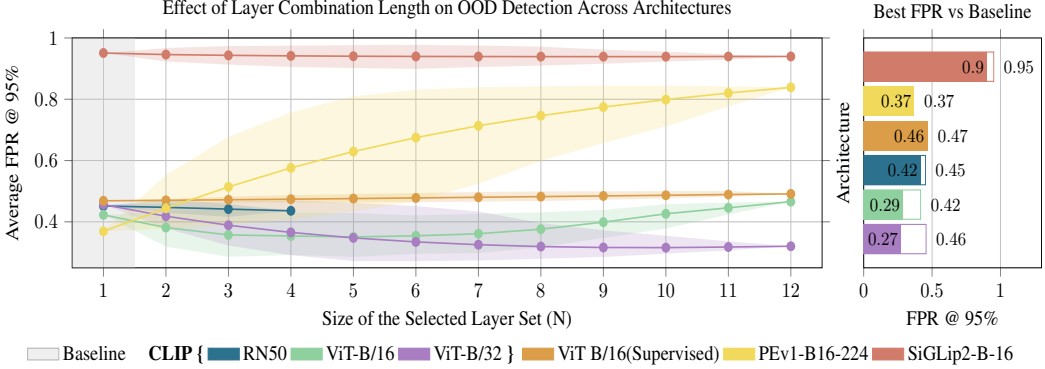

Figure 1: **Effect of layer combination length on OOD detection across architectures.** The line plot (left) shows average FPR@95 as a function of fused layer count $N$, where all combinations include the final layer (e.g., $N=3$ may yield layers $\{1, 2, 11\}$). Shaded regions indicate the full range of FPR@95 values across combinations. The gray zone marks the baseline (single-layer) case. The bar plot (right) compares baseline FPR@95 (gray) with the best result from layer fusion (colored) for each model. See Appendix M.16 for more details.

ViTs [14]. We examine how semantic diversity, entropy structure, and prediction stability vary across depth and training paradigms.

Our findings reveal strong contrasts between architectures. CLIP models exhibit high inter-layer diversity, broad top-1 agreement, and smooth entropy transitions across depth—properties that enable stable and effective multi-layer fusion. In contrast, supervised and self-supervised ViTs often show abrupt shifts or redundancy across layers, limiting the utility of intermediate feature fusion. Notably, more recent contrastive models such as SIGLIP and the Perception Encoder do not replicate CLIP's robustness, suggesting that architectural design plays a central role in enabling successful fusion strategies. As shown in Figure 1, combining three to six well-selected layers yields the largest performance gains, while fusing more layers tends to introduce redundancy or noise, diminishing returns. Additional analysis appears in Section 3.

Motivated by these insights, we introduce a simple yet effective extension of Maximum Concept Matching (MCM) [47]. Our method performs entropy-guided selection of informative intermediate layers and aggregates their outputs to improve OOD detection. It is fully training-free, inference-only, and does not require OOD data, fine-tuning, or prompt engineering. The method improves performance, reducing FPR@95—the false positive rate when the true positive rate is at 95%—by over 12 percentage points on far-OOD datasets and more than 7 points on near-OOD benchmarks (see Section 5). These results are consistent with the ablations presented in Section 6.

Extensive ablations confirm that our method is robust to hyperparameters such as fusion length, histogram resolution, and temperature scaling. We also observe that the best-performing layer combinations are consistent within a dataset, but differ across domains, emphasizing the need for dataset-adaptive strategies. These results challenge the assumption that the final layer is always optimal for OOD detection. By selectively leveraging signals from across the network, our approach offers a practical and general enhancement to training-free OOD methods.

**Our contributions are as follows:**

- We conduct a systematic analysis of intermediate representations across seven vision backbones and three training paradigms. While early layers alone are weak, aggregating selected intermediate layers consistently improves OOD robustness (Section 3).

- We propose a training-free extension of MCM that uses entropy-guided selection to identify and fuse informative layers. The method remains training-free and architecture-agnostic for CLIP (Section 4).

- We validate our approach across six OOD benchmarks and two ID datasets (ImageNet-1K and Pascal-VOC), achieving consistent gains in both single-label and multi-label settings. Our method outperforms prior inference-only baselines, particularly on contrastive architectures (Section 5).

## 2 Preliminaries

**OOD Detection.** Out-of-distribution (OOD) detection aims to identify test-time inputs drawn from a distribution different from the training one. Classical methods define a scoring function $G(\mathbf{x}) : \mathcal{X} \to \mathbb{R}$ using final-layer outputs of models pretrained on the in-distribution (ID) data. These approaches rely on post hoc confidence scores—e.g., softmax, energy, or entropy— overlooking earlier representations that may encode complementary semantic cues.

**Zero-shot OOD Detection with Pretrained Vision–Language Models.** Recent advances in large-scale contrastive vision–language models (e.g., CLIP [58]) have enabled zero-shot classification by aligning images and text within a shared embedding space. These models bypass conventional supervised training by leveraging natural language supervision for OOD detection. Given an ID label set $\mathcal{Y} = \{y_1, \ldots, y_K\}$, class-specific text prompts such as "a photo of a $y_i$" are embedded via a text encoder $T(y_i)$, while images are encoded through a visual encoder $I(x)$. Classification is then performed via cosine similarity between modalities:

$$p(y \mid x) = \frac{\exp\big(\cos(I(x), T(y)) \, / \, \tau\big)}{\sum_{y'} \exp\big(\cos(I(x), T(y')) \, / \, \tau\big)} \tag{1}$$

where $\cos(\cdot, \cdot)$ is the cosine similarity and $\tau$ a temperature parameter.

We use the Maximum Concept Matching (MCM [47]) approach to OOD detection, which flags an example as OOD if $\max_y p(y \mid x) < \theta$, with $\theta$ controlling sensitivity.

## 3 Exploring the Value of Intermediate Layers for OOD Detection

We investigate whether leveraging intermediate-layer representations—rather than relying solely on final outputs—can improve OOD detection in the zero-shot setting. To assess the potential of these representations, we extract features from intermediate layers throughout the network and evaluate their predictive utility for OOD detection. Specifically, let $\phi_v$ denote the visual encoder of a given model $a$. For each architecture, we obtain intermediate representations $\{\phi_v^{(\ell)}\}$ from a predefined set of layers,[1] allowing us to analyze how OOD-relevant information emerges and evolves throughout the network. Our analysis is conducted on ImageNet-1K as the in-distribution dataset, along with four diverse OOD datasets: iNaturalist [28], Places [75], SUN [69], and Textures [10]. Further details can be found in Appendix D.

For classification models, we apply the native classification head $h$ to each intermediate representation $\phi_v^{(\ell)}$. For contrastive models lacking such heads, we normalize $\phi_v^{(\ell)}$ as done at the final layer and apply the pretrained projection head $p$. To address layer-wise dimensionality mismatches—especially in ResNets—we insert a fixed random linear projection before applying $p$ (see Appendix I).

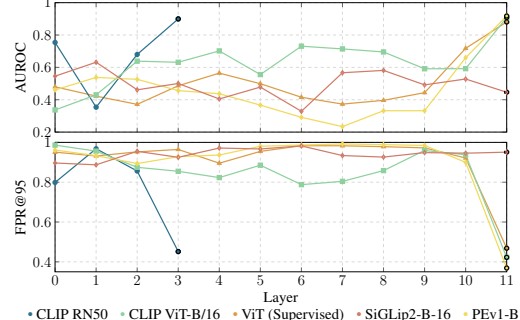

Importantly, this procedure is entirely inference-only: no new adapters or layers are introduced, no training phase is involved, and no in-distribution labels are used. All components are reused from the pretrained network. As a result, for each layer $\ell$, we obtain a class proba-

Figure 2: Layer-wise OOD detection performance across architectures. Most architectures exhibit their best performance near the final layer, while early layers generally under-perform.

bility distribution computed as $h(\phi_v^{(\ell)}(x))$, where $x$ is the input image and $\phi_v^{(\ell)}(x)$ is the intermediate representation at layer $\ell$.

To compute OOD scores, we use confidence-based indicators. For classification models, we extract the maximum softmax probability per layer (MSP) [25]. For contrastive models, we apply Maximum Concept Matching (MCM) [48], which computes softmax-normalized cosine similarities between

---

[1]Layer definitions follow standard naming and indexing of the Hugging Face Transformers library [68].

image and class embeddings. This yields a score tensor $S \in \mathbb{R}^{N \times L \times C}$, where $N$ is the number of input samples, $L$ the number of layers, and $C$ the number of classes. Each entry $S_{n,\ell,c}$ denotes the class-$c$ probability at layer $\ell$ for input $n$.

**Impact of a Single Layer for OOD Detection**   We begin by assessing the performance of OOD detection using features of individual layers $\ell$ in various architectures. As shown in Figure 2, most models reach their best AUROC and lowest FPR@95 at the final layer, highlighting the discriminative power of deeper representations. For instance, CLIP ViT-B/16 and PEViT-B16 show a consistent rise in AUROC with depth. Some intermediate layers offer isolated improvements, such as early gains in CLIP RN50 or reduced FPR@95 at layer 3 in DINOv2, but these cases are infrequent and strongly depend on the architecture. Similar trends are observed in MAE and MoCo v3, where mid-level layers provide only minor advantages; see Appendix M.17 for details.

> *Thus, relying on a single intermediate layer for OOD detection yields, at best, marginal and inconsistent improvements over the final layer, which remains the most reliable source of discriminative information.*

**Impact of Multi-Layer Fusion for OOD Detection**   Given the limited gains from single-layer prediction, we explore whether combining multiple layers can improve OOD detection. To this end, we aggregate OOD scores from selected subsets of intermediate layers, always including the final layer due to its strong baseline performance. These subsets span all layers and include all possible combinations for a given subset size.

Let $S \in \mathbb{R}^{N \times L}$ denote the matrix of confidence-based OOD scores, where $S_{n,\ell}$ is the OOD scores for the instance $n$ at layer $\ell$. Given a subset of layers $\mathcal{L} \subseteq \{1, \ldots, L\}$, we computed a fused score for each instance by averaging over the selected layers:

$$\bar{p} \in \mathbb{R}^{N \times 1}, \quad \bar{p}_n = \frac{1}{|\mathcal{L}|} \sum_{\ell \in \mathcal{L}} S_{n,\ell}. \tag{2}$$

This procedure yields a single fused OOD score $\bar{p}_n$ per instance. Figure 1 shows the average false positive rate (FPR) at $95\%$ as a function of the number of layers used in the fusion. Results are averaged over four OOD benchmarks—SUN, Places, Textures, and iNaturalist— with shaded regions representing the full range of FPRs (minimum to maximum) observed across all layer combinations of equal size. The plot reveals substantial differences in the FPR metric across architectures, with some models—such as SiGLip2 [61] and DINOv2 [54]— starting with notably higher FPRs, highlighting variability in baseline OOD performance.

Fusion improves performance in model-specific ways. CLIP variants show consistent FPR reduction as layers are added. MoCoV3 and MAE gain moderality, while supervised ViTs sees minimal change. SiGLIp2 and DINOv2 also improve, though less reliably. In contrast, larger models like CLIP ViT-L/14 and PEv1-B16-224 exhibit rising FPR with longer combinations, suggesting that aggregating too many layers may hurt performance when predictions conflict or lack discrimination.

While fusing multiple layers can enhance OOD detection, the specific number and choice of layers that contribute to this improvement vary across models and datasets. This raises the question of whether certain layers are consistently more informative. In Appendix E, we show that top-performing layer combinations tend to be structurally consistent within a given reference ID dataset, yet diverge significantly across datasets. These findings suggest that fusion strategies must be adapted to the domain, as no universal layer subset generalizes well across distributional shifts.

> *These results highlight the potential of multi-layer fusion for OOD detection while raising key questions: **what makes certain combinations effective, and how do they help?***

**Uncovering the Diversity of Intermediate Layers**   To better understand what makes certain layer combinations effective, we analyze the representational diversity of intermediate layers across models. We use SVCCA [59] to quantify structural similarity between layers by measuring the alignment of their activation patterns. As shown in Figure 3, contrastive models like CLIP exhibit a sharp decline in similarity as the layer distance $\Delta$ increases, indicating high inter-layer representational diversity and more progressive transformations. In contrast, supervised models and hybrid architectures display

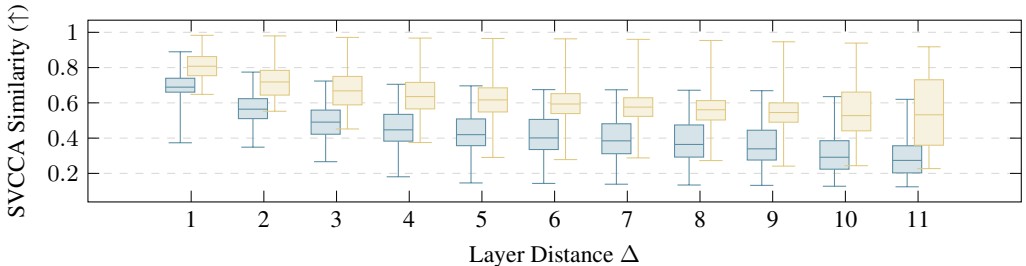

Figure 3: Layer-wise SVCCA similarity as a function of layer distance $\Delta$ for contrastive and classic models. CLIP models (blue) exhibit lower similarity across layers, indicating more progressive transformations, while Supervised models (yellow) retain higher layer-wise redundancy.

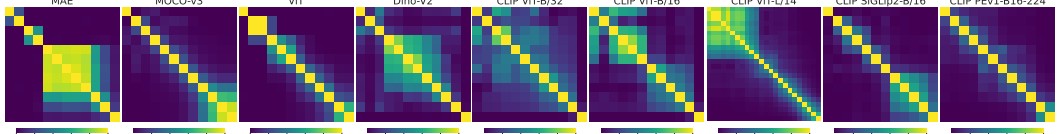

Figure 4: **Top-1 agreement similarity across transformer layers for various vision models.** Each matrix shows pairwise agreement in predicted top-1 classes across layers.

slower SVCCA decay, suggesting higher redundancy and smoother transitions between layers. These models tend to refine their representations gradually, concentrating most semantic abstraction near the final layers. By comparison, contrastive models distribute semantic information more evenly across depth, yielding more diverse intermediate features that may offer greater complementarity for fusion.

**Layerwise Prediction Agreement and Entropy** To complement the SVCCA analysis, we measure top-1 agreement across layers to assess prediction consistency. As shown in Figure 4, CLIP models—especially ViT-B variants—exhibit broad zones of agreement, reflecting stable prediction behavior across depth. This stability diminishes in larger models like ViT-L/14. Other contrastive models, such as SiGLip2 and PE, show lower agreement, suggesting abrupt shifts in prediction space. Supervised and self-supervised models (e.g., ViT, MoCo v3) display strong diagonals in their agreement matrices, suggesting independent prediction evolution across layers and limited stability, which can undermine the effectiveness of intermediate fusion.

To further contextualize these observations, Figure M.18 shows the average class entropy across layers. Distinct calibration patterns emerge: supervised models and MAE exhibit sharp entropy drops near the final layers, indicating increasingly confident—often overconfident—predictions [22, 46]. Earlier layers, by contrast, retain high entropy with near-uniform class distributions, offering little semantic guidance This weak discriminative signal can limit their contribution in feature aggregation. In contrast, CLIP models display a more gradual change in entropy across depth, with moderate uncertainty in early layers and more confident predictions in deeper ones. This structured progression allows intermediate layers to contribute non-redundant yet coherent information. Conversely, models like SiGLip2 maintain uniformly high entropy, implying consistently flat softmax outputs that limit the effectiveness of multi-layer fusion. Perception Encoder [4] shows a similar issue, with little differentiation between earlier and later layers.

> *These trends provide an explanatory basis for the differing effectiveness of intermediate-layer fusion observed across architectures. **Only architectures that balance inter-layer diversity and prediction consistency**—most notably CLIP—derive substantial benefit from multi-layer fusion. In contrast, models with unstable or flat prediction profiles across depth may require more selective or weighted aggregation strategies to avoid destructive interference.*

## 4 Proposed Method to Exploit Intermediate-Layer Representations

Motivated by the patterns uncovered in our analysis of intermediate-layer fusion, Figure 5 presents our training-free OOD detection framework $G(\cdot)$, which leverages intermediate-layer representations

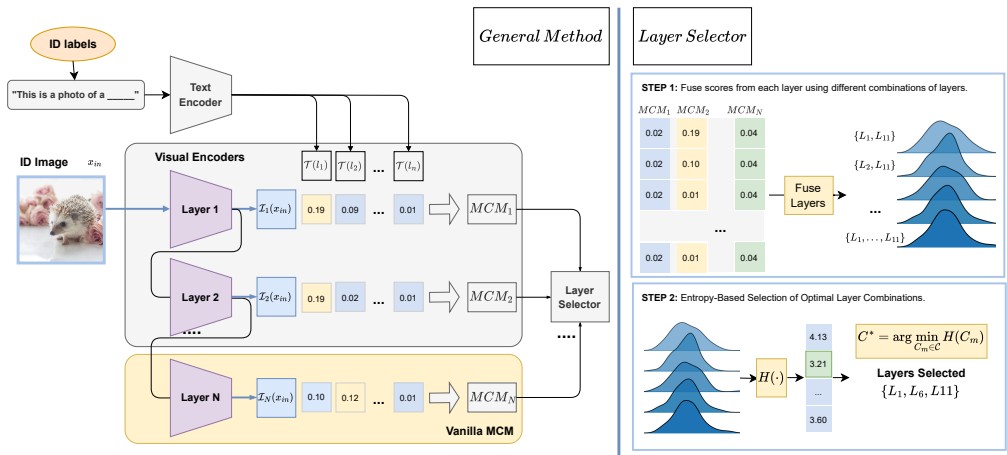

Figure 5: We propose a general approach to OOD detection that exploits features from intermediate layers of a visual encoder (left), extending the *Maximum Concept Matching* (MCM) method [47]. Section 3 analyzes the informativeness of intermediate features across architectures. Based on these insights, Section 4 introduces an entropy-based layer selector (right) that identifies the most reliable combination of layers for training-free OOD detection.

from pretrained vision–language models. We extend the Maximum Concept Matching (MCM) method across multiple layers and introduce an entropy-based layer selection mechanism that identifies the most discriminative combinations for OOD detection.

**Text and Visual Encodings.** Each in-distribution (ID) class label $y_i \in \mathcal{Y}_{\text{in}}$ is mapped to a text prompt (e.g., "a photo of a $y_i$") and encoded into a semantic embedding $T(t_i)$ using a pretrained text encoder. On the visual side, backbones such as ViT or ResNet yield intermediate feature representations $\{\phi^{(1)}(x), \ldots, \phi^{(L)}(x)\}$. Each feature $\phi^{(\ell)}(x)$ is mapped into a shared embedding space via a projection head $p$, yielding $I_\ell(x) = p(\phi^{(\ell)}(x))$. This enables alignment between visual and textual modalities for similarity-based matching.

**Maximum Concept Matching (MCM).** For each test input $x'$, we compute cosine similarities between the visual embedding $I_\ell(x')$ and all text embeddings $\{T(t_k)\}_{k=1}^K$. The MCM score at layer $\ell$ is defined as:

$$S_{\text{MCM}}^{(\ell)}(x') = \max_j \frac{\exp\left(\cos\left(I_\ell(x'), T(t_j)\right)/\tau\right)}{\sum_{k=1}^K \exp\left(\cos\left(I_\ell(x'), T(t_k)\right)/\tau\right)}, \tag{3}$$

where $\tau$ is a temperature parameter that controls the sharpness of the distribution. This normalized score approximates the model's confidence at each layer.

**Entropy-Based Layer Selection.** To aggregate information across layers, we define a set of candidate combinations $\mathcal{C} = \{C_1, \ldots, C_M\}$, where each $C_m \subseteq \{1, \ldots, L\}$ always includes the final layer. For each combination $C_m$, we compute the aggregated score:

$$S_{\text{MCM}}(x'; C_m) = \frac{1}{|C_m|} \sum_{\ell \in C_m} S_{\text{MCM}}^{(\ell)}(x'). \tag{4}$$

To evaluate each combination $C_m$, we compute the distribution $\{S_{\text{MCM}}(x_i'; C_m)\}_{i=1}^N$ over a set of unlabeled in-distribution samples $\{x_i'\}_{i=1}^N$, and bin the scores into a normalized histogram $\{p_b\}_{b=1}^B$. We then compute the entropy:

$$H(C_m) = -\sum_{b=1}^B p_b \log(p_b), \tag{5}$$

A low-entropy distribution indicates confident and concentrated scores, which we use as a proxy for better ID–OOD separability. The optimal layer combination is then selected as:

$$C^* = \arg\min_{C_m \in \mathcal{C}} H(C_m). \tag{6}$$

Table 1: **Training-free OOD detection results on ImageNet-1K and Pascal-VOC as ID datasets.** We report the false positive rate at 95% TPR (FPR95, ↓) and AUROC (↑) across six OOD datasets: iNaturalist, SUN, Places, DTD, ImageNet-22K, and COCO. Best results per column are highlighted in bold.

| Method | Backbone | iNaturalist | | SUN | | Places | | Texture | | ImageNet22K | | COCO | | Average | |
|---|---|---|---|---|---|---|---|---|---|---|---|---|---|---|---|
| | | FPR↓ | AUC↑ | FPR↓ | AUC↑ | FPR↓ | AUC↑ | FPR↓ | AUC↑ | FPR↓ | AUC↑ | FPR↓ | AUC↑ | FPR↓ | AUC↑ |
| **ID - ImageNet1K** | | | | | | | | | | | | | | | |
| MCM | ViT-B/32 | 33.85 | 93.62 | 40.99 | 91.56 | 46.71 | 89.25 | 60.90 | 85.03 | - | - | - | - | 45.61 | 89.87 |
| MCM | ViT-B/16 | 30.67 | 94.63 | 37.41 | 92.57 | 43.67 | 89.96 | 57.34 | 86.18 | - | - | - | - | 42.27 | 90.83 |
| **SeTAR + MCM** | ViT-B/16 | 26.92 | 94.67 | 35.57 | 92.79 | 42.64 | 90.16 | 55.83 | 86.58 | - | - | - | - | 40.24 | 91.05 |
| GL-MCM | ViT-B/16 | 17.42 | 96.44 | 30.75 | **93.44** | 37.62 | 90.63 | 55.20 | 85.54 | - | - | - | - | 35.25 | 91.51 |
| **SeTAR + GL-MCM** | ViT-B/16 | 13.36 | 96.92 | **28.17** | 93.36 | 36.80 | 90.40 | 54.17 | 84.59 | - | - | - | - | 33.12 | 91.32 |
| **Ours** | ViT-B/16 | 15.98 | 96.90 | 45.58 | 89.69 | 35.71 | **92.72** | 25.51 | **94.84** | - | - | - | - | 30.70 | **93.54** |
| **Ours** | ViT-B/32 | **12.02** | **97.64** | 28.98 | **93.37** | 35.69 | 91.61 | 39.36 | 91.07 | - | - | - | - | **29.01** | 93.42 |
| **ID - Pascal-VOC** | | | | | | | | | | | | | | | |
| MCM | ViT-B/32 | 34.80 | 95.35 | 30.60 | 93.74 | 37.70 | 91.99 | 51.60 | 91.68 | 55.00 | 91.16 | 59.10 | 89.23 | 44.80 | 92.19 |
| MCM | ViT-B/16 | 10.51 | 97.93 | 30.45 | 94.25 | 36.11 | 91.86 | 53.21 | 91.77 | 53.82 | 91.12 | 57.10 | 89.02 | 40.20 | 92.66 |
| **SeTAR + MCM** | ViT-B/16 | 4.38 | 98.70 | 26.24 | 94.95 | 28.67 | 93.28 | 50.32 | 92.32 | 44.61 | 92.63 | 49.80 | 89.68 | 34.00 | 93.59 |
| GL-MCM | ViT-B/16 | 4.33 | 98.81 | 22.94 | 94.63 | 26.20 | 93.11 | 41.61 | 92.88 | 37.88 | 93.17 | 43.70 | 90.71 | 29.44 | 93.88 |
| **SeTAR + GL-MCM** | ViT-B/16 | 3.01 | **99.04** | 21.76 | 94.98 | 24.00 | 93.73 | 37.61 | 93.87 | **33.46** | **94.24** | 40.60 | **91.48** | **26.74** | **94.56** |
| **Ours** | ViT-B/32 | 32.18 | 94.40 | 35.60 | 92.61 | 49.17 | 92.34 | **33.72** | **95.78** | 54.36 | 91.34 | 57.60 | 89.97 | 43.77 | 92.74 |
| **Ours** | ViT-B/16 | **2.19** | 98.92 | **19.70** | **95.77** | **19.53** | **95.11** | 44.08 | 92.06 | 34.70 | 91.15 | 41.60 | 88.09 | 26.97 | 93.52 |

**Inference.** At test time, the selected combination $C^*$ is used to compute the final OOD score for any input $x'$ using Eq. (4). This enables robust prediction by fusing complementary information across informative layers.

## 5 Experiments

### 5.1 Dataset and Experimental Settings.

We use two ID datasets: **ImageNet-1K** [12], a standard large-scale classification benchmark, and **Pascal-VOC** [15], a multi-object detection dataset. Following prior work [30, 51]. Detailed dataset descriptions and statistics are provided in Appendix D. For ImageNet-1K, we use **iNaturalist** [28], **SUN** [69], **Places** [75], and **Textures** [9] as OOD datasets, following existing protocols exposed in [30]. For Pascal-VOC, we use **ImageNet-22K** [60], **iNaturalist**, **SUN**, **Textures** and we add **MS-COCO** [42] as an auxiliary OOD source when Pascal-VOC is used as the ID dataset.

**Models and Setup.** We use **CLIP** [58] with a ViT-B/16 backbone as our primary contrastive vision model. All evaluations are performed in a fully training-free setting without any additional fine-tuning. To ensure comparability, we apply the softmax function with a fixed temperature $\tau = 1$. For consistency and to reduce variance in the fusion process, we fix the number of combined layers to $L \leq 5$ (see 6). Our experimental setup uses a single NVIDIA RTX 4090 GPU. The selection of the optimal layer combination takes approximately 50 seconds for both ImageNet-ID and Pascal-VOC.

**Evaluation Metrics and Baselines.** We report (i) **FPR@95**: the false positive rate when the ID true positive rate is 95% (lower is better), and (ii) **AUROC**: the area under the ROC curve for distinguishing ID and OOD samples (higher is better). We compare our method against recent training-free, zero-shot baselines, including **MCM** [48], **GL-MCM** [51], and **SETAR** [40].

### 5.2 Results

Table 5.1 presents training-free OOD detection results across six benchmarks, using ImageNet-1K or Pascal-VOC as ID datasets. Our method achieves the best average performance in both settings. With ViT-B/16, we obtain an FPR95 of **30.70%** and AUROC of **93.54** on ImageNet-1K, and **26.97%** for the FPR95 and **93.52** for the AUROC on Pascal-VOC. Gains are consistent across iNaturalist, Texture, COCO, and ImageNet-22K, which span fine-grained and complex domains. The largest improvements occur on COCO and Texture, where standard ID/OOD assumptions often break. ViT-B/32 yields smaller gains on Pascal-VOC, likely due to its lower spatial resolution, echoing prior findings from GL-MCM and SeTAR that emphasize the importance of localized features. We provide further insight into the semantic abstraction patterns driving these improvements in F.

Unlike MCM, which uses only the final-layer `[CLS]` token, our method aggregates selected intermediate layers. On ImageNet-1K with ViT-B/16, this reduces FPR95 by **51.9%** relative on iNaturalist, achieves **55.5%** relative reduction on Texture, and **18.2%** on Places. On Pascal-VOC with ViT-B/16, relative reductions of **79.2%** on iNaturalist, **35.3%** on SUN, **45.9%** on Places, and **27.1%** on COCO are observed. These gains are most pronounced in high-variability settings, where final-layer predictions tend to be overconfident.

**Near OOD is better with intermediate layer** We evaluate near-OOD performance on the NINCO and SSB-Hard benchmarks, following the protocol introduced in [73]. As shown in Table 2, leveraging intermediate-layer representations yields consistent improvements across both datasets. Our method outperforms the training-free baseline (MCM) by over **7.5%** in AUROC and reduces the false positive rate by more than 10 points. Com-

Table 2: **Near-OOD results.** Best results per column are bolded.

| Method | NINCO | | SSB-Hard | | Average | |
|---|---|---|---|---|---|---|
| | AUC↑ | FPR↓ | AUC↑ | FPR↓ | AUC↑ | FPR↓ |
| MCM | 73.53 | 79.69 | 62.71 | 90.47 | 68.12 | 85.08 |
| GL-MCM | 76.03 | 74.35 | 66.13 | 87.42 | 71.08 | 80.885 |
| SeTAR | 76.95 | 70.16 | 68.31 | 84.94 | 72.63 | 77.535 |
| LoCoOp | 73.90 | 80.04 | 65.73 | 89.48 | 69.81 | 84.76 |
| SeTAR+FT | 77.56 | 70.86 | 69.68 | 83.99 | 73.62 | 77.425 |
| Ours | **80.98** | **66.74** | **70.44** | **82.80** | **75.71** | **74.77** |

pared to fine-tuned approaches such as SETAR+FT, our method achieves a **1.92%** lower FPR while maintaining competitive AUROC. These results indicate that the inductive biases encoded in intermediate-layer representations provide a more robust foundation for handling subtle distribution shifts, especially in near-OOD regimes, than fine-tuning on in-distribution data alone.

**Different architectures and prompt-based methods** To test portability across backbones and prompting strategies (Table 3), we find intermediate-layer fusion remains effective beyond the default setting. Unlike single-layer baselines, our method aggregates selected intermediate layers. With MCM on RN50x4, we obtain an FPR95 of **41.61%** and AUROC of **90.80**, while ViT-L/14 achieves **37.09%** for the FPR95 and **91.70** for the AUROC. Prompted ViT-B/16 also benefits substantially: NegLabel reaches **23.79%** FPR95 and **95.05** AUROC, while CSP achieves **15.52%** for the FPR95 and **96.74** for the AUROC. These trends mirror our layer-agreement analysis: heterogeneous ResNet stages let fusion reduce variance and suppress spurious peaks, whereas ViT-L/14's higher inter-layer redundancy and sharper

Table 3: **Intermediate vs last layer performance improvement across scoring rules.**

| Method | Average | |
|---|---|---|
| | FPR↓ | AUC↑ |
| *Extra backbones (MCM)* | | |
| MCM (Last Layer) - RN50x4 | 45.16 | 89.95 |
| MCM (Intermediate) - RN50x4 | **41.61** | **90.80** |
| MCM (Last Layer) - ViT-L/14 | 37.16 | 91.66 |
| MCM (Intermediate) - ViT-L/14 | **37.09** | **91.70** |
| *Prompt methods (composition with ViT-B/16)* | | |
| NegLabel | 25.40 | 94.21 |
| NegLabel + Int. Layers | **23.79** | **95.05** |
| CSP | 17.51 | 95.76 |
| CSP + Int. Layers | **15.52** | **96.74** |

softmax (with $\tau = 0.01$) leave less headroom. Dataset context matters J: the largest FPR drops appear on scene-centric SUN/Places where early layers retain layout/background cues; gains are smaller on DTD and iNaturalist, where final embeddings already align with object-centric semantics. Overall, fusion acts at the representation level and transfers robustly across architectures and prompts.

**Orthogonality to Scoring Rules** To evaluate whether our contribution acts at the representation level rather than tailoring a specific decision rule, we report results across multiple OOD scores (Table 4). The considered scores capture complementary statistics: peak sensitivity for MaxLogit [25] and MCM [47], and global aggregation for Energy [44], Entropy, and Variance [47]. Our method demonstrates consistent improvements across all scoring functions and architectures, with particularly strong gains on Vision Transformers where distributional

Table 4: **Intermediate vs last layer performance improvement across scoring rules.** Results averaged across SUN, Places365, DTD, and iNaturalist datasets.

| Method | ResNet | | ViT-B/32 | | ViT-B/16 | | Average | |
|---|---|---|---|---|---|---|---|---|
| | AUC↑ | FPR↑ | AUC↑ | FPR↑ | AUC↑ | FPR↑ | AUC↑ | FPR↑ |
| *Distributional Methods* | | | | | | | | |
| Entropy | +0.1 | +0.0 | +1.1 | +2.5 | +9.0 | +16.0 | +3.4 | +6.2 |
| Energy | +0.5 | +0.2 | +4.2 | +0.4 | +2.4 | +0.6 | +2.4 | +0.4 |
| Variance | +0.1 | +0.1 | +1.0 | +2.3 | +8.9 | +16.0 | +3.3 | +6.1 |
| *Max-based Methods* | | | | | | | | |
| MaxLogit | +1.2 | +4.0 | -0.9 | +0.3 | +3.2 | +8.6 | +1.2 | +4.3 |
| MCM | +0.8 | +3.5 | +3.6 | +16.6 | +2.7 | +11.6 | +2.4 | +10.2 |

methods benefit most from intermediate-layer fusion (More details in K). These consistent im-

provements span both peak-based and aggregation-based metrics, ruling out score-specific artifacts and ensuring compatibility with pipelines that already fix a scoring rule. The consistency across this diverse family of scores confirms that intermediate-layer fusion provides broadly applicable representation improvements rather than optimizing for a particular decision boundary.

# 6 Discussion

**Computational Cost Analysis of Intermediate Layer Usage** We evaluate the computational implications of incorporating intermediate layer features across three CLIP architectures: RN50x4, ViT-B/16, and ViT-B/32. The study varies batch size from 1 to 8 and reports peak GPU memory, per-image latency, total inference time, and throughput (images per second). As shown

Table 5: **Efficiency of intermediate-layer fusion across backbones and batch sizes.**

| Architecture | Memory (GB) | | Latency (ms) | |
|---|---|---|---|---|
| | Last↓ | All↓ | Last↓ | All↓ |
| ResNet-50x4 | 0.41±0.00 | 0.45±0.04 | 67.70±60.43 | 73.80±63.52 |
| ViT-B/16 | 0.57±0.00 | 0.63±0.05 | 218.47±154.88 | 224.96±184.71 |
| ViT-B/32 | 0.57±0.01 | 0.59±0.02 | 211.46±175.48 | 219.46±167.30 |

in Table 5 and Figures L.13, L.15, and L.14, extracting intermediate features adds only modest overhead. Across all configurations, memory and latency remain within practical deployment limits, with minimal increases relative to the last-layer baseline. The effect is smallest for transformer architectures, which sustain competitive throughput. Full details appear in Appendix L.

**Ablation: Layer Selection Strategies.** We evaluate the effect of different heuristics for selecting intermediate layers to fuse in training-free OOD detection with CLIP ViT-B/16. Since evaluating all possible combinations is computationally infeasible, we consider seven strategies based on per-layer output statistics: entropy, kurtosis, standard deviation, Gini coefficient, Jensen–Shannon (JSD) divergence, average scoring across all layers, and random selection. Each method selects up to $L = 5$ layers using softmax-normalized logits at a fixed temperature $\tau = 1$.

These metrics are chosen for their sensitivity to the *shape* of the class probability distribution. Our hypothesis is that layers with peaked, asymmetric distributions are more effective for OOD separation, while flatter distributions yield less discriminative scores. For example, the **Gini coefficient** $(1 - \sum_i p_i^2)$ increases with concentration, while **JSD divergence** measures deviation from uniformity in a bounded, symmetric form. The *average* baseline aggregates scores from all layers without selection, serving as a naive fusion strategy.

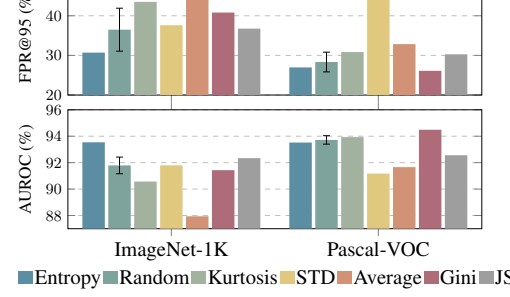

Figure 6: **Ablation of layer selection strategies.** Comparison of selection heuristics.

Figure 6 reports FPR@95 and AUROC on ImageNet-1K and Pascal-VOC. Entropy-based selection consistently outperforms all other methods, achieving the lowest false positive rates and highest AUROC. In contrast, statistical heuristics such as kurtosis and standard deviation yield less reliable results. Although random selection performs similarly on average, its high variance (omitted for clarity) highlights the advantage of informed scoring.

**Ablation: Sensitivity to Histogram Binning** ($B$)**, Combination Length** ($L$)**, and Temperature** ($\tau$) We evaluate the sensitivity of our method to three key hyperparameters: the number of histogram bins used for score normalization, the combination length (i.e., the number of layers fused), and the temperature applied to softmax outputs. Overall, the method exhibits stable performance across a broad range of configurations. For histogram binning, AUROC and FPR@95 remain consistent when using between 16 and 128 bins, with noticeable degradation outside this range due to under- or over-discretization effects. In the case of combination length, fusing 3 to 6 layers yields optimal results, capturing sufficient semantic diversity without introducing excessive redundancy (see G for a more detailed analysis); shorter combinations lead to marginal improvements, while longer ones tend to incorporate noisy or less informative layers, reducing detection reliability. Finally, temperature scaling demonstrates that values in the range $[0.25, 2.0]$ preserve stable behavior, whereas values

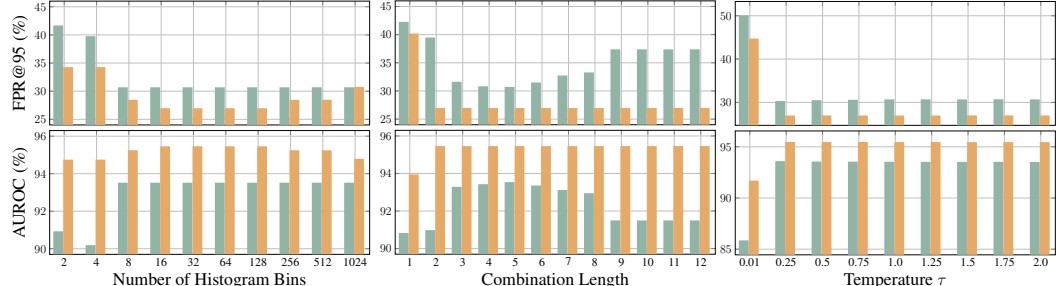

Figure 7: Ablation studies across different hyperparameter dimensions: number of histogram bins (left), combination length (center), and temperature $\tau$ (right). ■ ImageNet-1K ■ Pascal-VOC.

below 0.25 cause a sharp decline in performance, likely due to overconfident predictions. These findings confirm that our method is robust to hyperparameter choices within the empirically validated ranges reported above.

# 7 Related Work

**Out-of-distribution detection.** OOD detection identifies inputs that deviate from the training distribution, a key challenge for safe deployment. Early methods used softmax-based confidence scores, such as MSP and ODIN [25, 41], or statistical distances like Mahalanobis [37]. Later work introduced energy-based scores [44] and Outlier Exposure (OE) [27] to better separate ID and OOD data. Other advances explore residual features [66], hyperspherical embeddings [49], and activation shaping [13]. Theoretical studies have analyzed learnability and scoring foundations [3, 16, 52], while modality-specific taxonomies [1] extend applicability. In NLP, OOD work focuses on intent classification [72], data augmentation [6], uncertainty [29], and unsupervised adaptation [70], often using pretrained transformers [26, 56].

**Vision-language models for OOD detection.** VLMs like CLIP [57] support zero-shot OOD detection by aligning image and text embeddings. Recent extensions explore concept matching [47, 51], prompt tuning [50, 39], and synthetic outliers [5]. Training-free variants such as SeTAR [40] further expand this space. Broader efforts leverage contrastive learning [63, 65], noisy supervision [33], and models like ViLT, VisualBERT, and Tip-Adapter [35, 38, 74]. Studies on calibration and robustness [67, 19] highlight strengths and weaknesses under distribution shift. While prior work focuses on final-layer embeddings, we show that intermediate CLIP layers offer improved detection, influenced by depth and patch size.

**Intermediate-layer representations.** Early distance-based detectors used intermediate features [37], but most modern methods rely on final-layer outputs. Recent work shows that intermediate layers carry transferable, shift-stable signals [62, 32], typically with added supervision or retraining. Lin et al. [43] target closed-set classifiers with intermediate exits and exit-wise energy plus input-conditioned early stopping, not VLM embeddings. Fayyad et al. [18] add supervised auxiliary heads; Guglielmo and Masana [20] mainly select a single discriminative layer on supervised backbones and small-scale benchmarks rather than fuse across depth. In contrast, our approach is training-free and VLM-native, select complementary layers using simple ID-only statistics.

# 8 Conclusion

We challenge the final-layer paradigm in OOD detection by showing that intermediate representations, especially in contrastive models like CLIP, encode diverse and complementary signals. Our training-free method selects and fuses informative layers via entropy-based scoring, yielding consistent improvements across near- and far-OOD benchmarks. These results highlight the architectural differences that shape layer utility and demonstrate the value of exploiting internal model structure for robust, inference-only OOD detection.

**Acknowledgment** This work was supported by *Naval Group*. The authors are affiliated with the *Centre National de la Recherche Scientifique (CNRS)* and the *CROSSING Lab*.

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

# A   Limitations

Despite demonstrating consistent improvements, several limitations highlight promising directions for future research. First, while our method benefits from the structured and diverse intermediate representations found in architectures like CLIP, it remains less effective in models with flatter or redundant representations, such as MAE or Perception Encoder. Future work could explore adaptive or weighted fusion strategies to mitigate these limitations. Second, although our entropy-based layer selection is inference-only and label-free, the selected combinations vary significantly across ID datasets. This domain sensitivity suggests the need for more generalizable selection mechanisms, potentially incorporating Outlier Exposure or few-shot adaptation. Third, our study is restricted to visual inputs. Extending the method to other modalities—such as audio or language—could further test the versatility of intermediate-layer fusion for OOD detection, especially in domains where semantic abstractions may emerge differently across depth.

# B   Ethical Considerations

This work proposes a training-free method for out-of-distribution (OOD) detection based on intermediate-layer fusion in pretrained vision-language models. The method does not involve the collection or use of personal data, nor does it introduce new generative components or learning from sensitive content. It operates entirely at inference time using publicly available models and datasets. As such, it poses minimal risk in terms of privacy, fairness, or misuse. Nonetheless, we acknowledge that robust OOD detection plays a vital role in the safe and ethical deployment of machine learning systems. Future research should continue to examine the implications of model biases and domain shifts to support fair and transparent AI behavior in real-world applications.

# C   Societal Impact

Improving OOD detection is crucial for the reliability of machine learning systems deployed in open-world environments, including safety-critical domains such as healthcare diagnostics, autonomous vehicles, and security applications. Our method contributes toward the development of AI systems that can identify and appropriately reject unfamiliar or anomalous inputs, thus preventing overconfident and potentially harmful predictions. Additionally, accurate OOD detection may support tasks such as active learning, human-in-the-loop filtering, and data selection for continual learning. By enabling more trustworthy model behavior under distributional shift, this line of research contributes to the broader goal of responsible and safe AI deployment.

# D   Datasets

Following prior work [30], we adopt the standardized configuration for in-distribution (ID) and out-of-distribution (OOD) datasets that has been widely used in recent CLIP-based OOD detection benchmarks. Specifically, we use ImageNet-1K [12] and Pascal-VOC [15] as ID datasets. For OOD evaluation, we follow the curated protocol proposed in MoS [30], which provides de-duplicated subsets of four OOD datasets: iNaturalist [28], SUN [69], Places [75], and Texture (DTD) [10]. Additional OOD sets include ImageNet-22K [60] and MS-COCO [42].

**In-Distribution Datasets**

**ImageNet1K**   ImageNet-1K [12] serves as our primary in-distribution dataset. Consistent with prior evaluations [30, 48], we assess OOD detection using the full 50,000-image validation split, which spans 1,000 object categories and is widely adopted for benchmarking zero-shot performance.

**Pascal-VOC**   For object-centric evaluation, we utilize Pascal-VOC [15] as an ID dataset. Following the setup introduced by Miyai et al. [51], we select samples containing only a single labeled object per image. To ensure comparability with SeTAR [40], we adopt the 906-image test set configuration used in their study.

**Out-of-Distribution Datasets**

**iNaturalist**   iNaturalist [28] is a large-scale dataset of flora and fauna images. We employ the 10,000-image version curated by Huang et al. [30], where overlapping classes with ImageNet-1K are removed to ensure distributional shift.

**Places**   The Places dataset [75] contains millions of labeled scenes across various environments. For evaluation, we rely on a 10,000-image subset filtered by [30] to eliminate semantic overlap with the ID categories.

**SUN**   SUN [69] features a broad collection of indoor and outdoor scene images. We use the filtered version compiled by [30], which retains 10,000 samples that are semantically distinct from ImageNet-1K.

**Texture (DTD)**   The Describable Textures Dataset [10] includes fine-grained texture patterns organized into 47 visual descriptors. In line with [30], we employ the full set of 5,640 images, which are disjoint from the object categories in our ID datasets.

**ImageNet22K**   ImageNet-22K [60] extends the standard ImageNet taxonomy to over 21,000 categories. We use the version filtered by Ming et al. [48], which excludes all classes overlapping with ImageNet-1K and Pascal-VOC, to serve as OOD input in both evaluation settings.

**MS-COCO**   MS-COCO [42] is used in a filtered setting (COCO-OOD) containing 1,000 samples that are class-wise disjoint from Pascal-VOC. We use the version from [48] to test OOD detection performance when Pascal-VOC is the ID distribution.

**Near-OOD Datasets**

We follow the recommendations from OpenOOD [73] to evaluate near-OOD detection using two challenging benchmarks that emphasize subtle distributional shifts while maintaining high visual similarity to in-distribution data.

**NINCO**   NINCO [24] is a curated benchmark for evaluating nuanced OOD detection in natural images. It includes near-OOD samples that are visually similar to ImageNet-1K categories but originate from distinct distributions.

**SSB-Hard**   SSB-Hard [64] introduced for open-set recognition on ImageNet-1K. It provides only OOD samples with minimal visual deviation but distinct semantics.

# E   Diversity and Consistency of Oracle Layer Combinations

We assess the internal diversity of oracle-selected layer combinations in Figure E.8 using the Jaccard distance [31], computed among the top-10 performing combinations. A lower Jaccard distance indicates stronger agreement in the selected layers across different top-performing combinations. As shown, both ImageNet and Pascal-VOC exhibit low pairwise distances within their top-10 selections (mean Jaccard distances of 0.480 and 0.477, respectively), suggesting high internal consistency. This implies that, once an ID dataset is fixed, the best-performing combinations tend to share many common layers, indicating that certain representations are consistently beneficial for OOD detection within each domain.

However, this consistency does not extend across datasets. While individual ID datasets demonstrate stable oracle combinations, the actual sets of selected layers differ substantially between ImageNet and Pascal-VOC. This observation is further supported by Figure E.9, which quantifies the overlap between top-$k$ layer selections across datasets. Notably, the intersection between the top-50 combinations of ImageNet and Pascal comprises only four shared configurations. These shared combinations yield an average FPR of 0.300, which is a worst fpr than the 0.288 average obtained by selecting layers independently for each ID dataset using an entropy-based heuristic. This discrepancy indicates that a single fixed set of layers is inadequate for generalizing across ID datasets, and that the optimal composition of layers for OOD detection is highly dataset-dependent.

> *These findings underscore two key insights: (1) within a given ID dataset, the top-performing layer combinations exhibit high structural consistency, and (2) across different ID datasets, the optimal configurations diverge significantly, reflecting distinct underlying representational demands. This analysis reinforces the conclusion that static fusion strategies are insufficient for general-purpose OOD detection.*

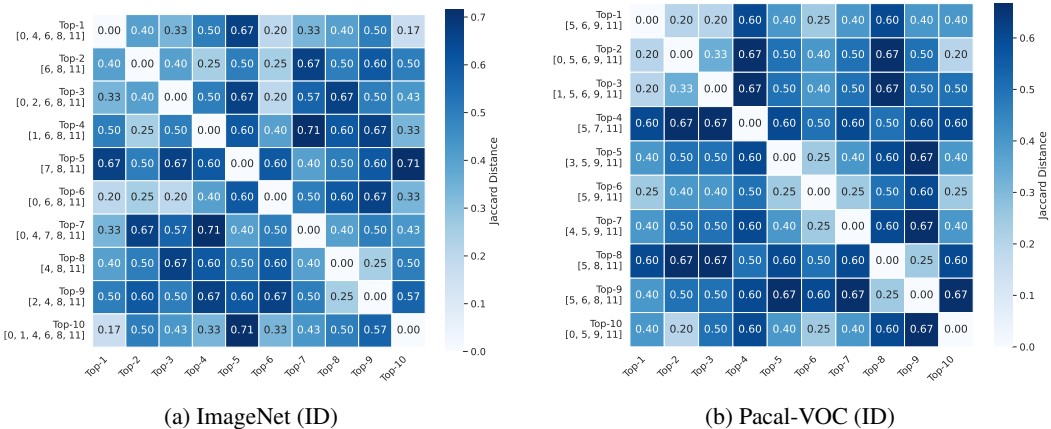

(a) ImageNet (ID)             (b) Pacal-VOC (ID)

Figure E.8: Jaccard distance between the top-10 oracle combinations under two different ID datasets. Lower values indicate higher agreement between selected layer sets.

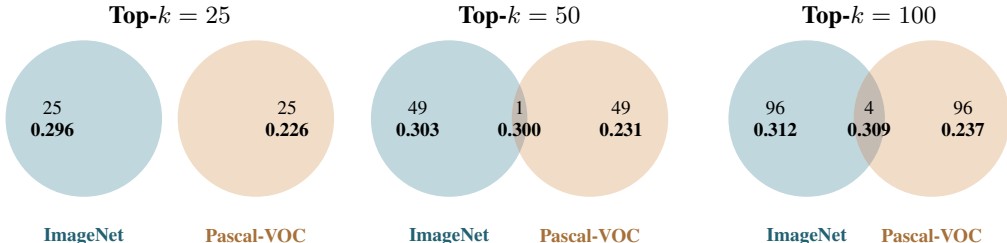

Figure E.9: Overlap of top-$k$ oracle layer combinations between ImageNet and Pascal-VOC. Intersection regions are shaded in gray and color-matched to the scatterplot. The bold decimal values indicate the average FPR computed across all OOD datasets evaluated under each ID configuration (ImageNet or Pascal-VOC).

# F   Semantic concept emergence across layers.

Figure F.10 offers an interpretive bridge between our theoretical foundations (Section H), representational analyses (Section 6), and empirical evaluation of intermediate-layer fusion strategies. It visualizes the layer-wise distribution of six high-level semantic categories, as defined by [53], across eight datasets. Each curve denotes the average proportion of concept-relevant descriptors matched at each transformer layer, thereby revealing both architectural regularities and dataset-specific semantic emergence.

This figure directly substantiates **Assumption A1 (Layered Semantics)** from Section H. Low-level concepts (e.g., *colors*, *textures*) consistently peak in early layers (typically layers 2 to 5), while high-level concepts (e.g., *objects and machines*, *activities*) peak in deeper layers (layers 9 to 12). This stratification reinforces the canonical coarse-to-fine information processing in vision transformers, but here it is quantified via *explicit semantic concept activation*, rather than inferred from feature abstraction or classification accuracy. These results confirm that semantic depth varies across layers, validating our use of intermediate-layer fusion to capture complementary features.

Moreover, Figure F.10 deepens the interpretation of the SVCCA-based similarity trends in Figure 3. As discussed in Section 6, contrastive models exhibit lower similarity across layers, indicative of

greater representational diversity. The semantic traces in Figure F.10 offer an explanation: layers specialize in distinct concept types. In particular, the delayed emergence of categories such as *objects* supports our hypothesis that last-layer-only approaches may overlook crucial intermediate semantics. This aligns with the performance gains we observe when fusing information across multiple layers.

Finally, we note that the concept distributions are remarkably consistent across datasets, suggesting that concept emergence is not purely data-driven but also reflects model-internal inductive biases. This insight supports the transferability of our layer selection heuristics: strategies optimized on one dataset (e.g., ImageNet) remain effective across others (e.g., COCO, SUN), as demonstrated in our zero-shot evaluations for OOD detection.

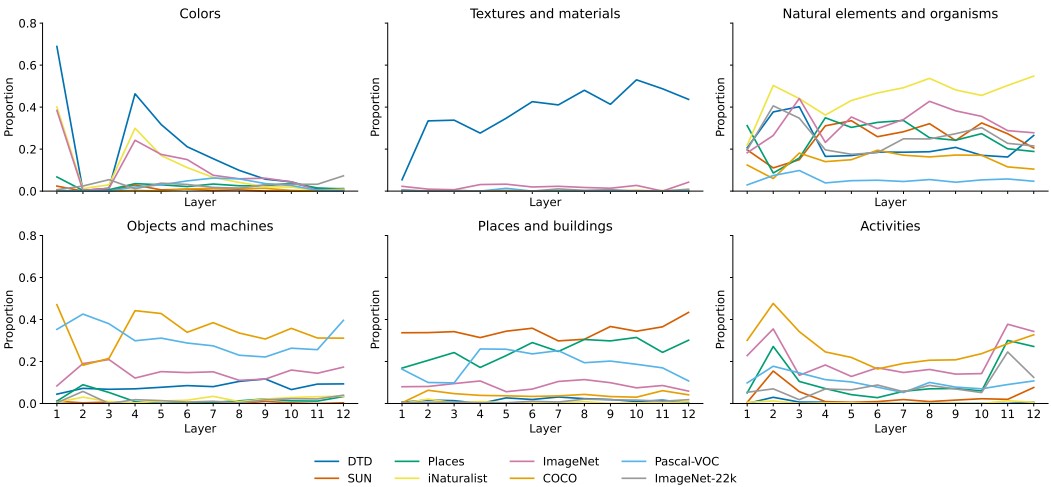

Figure F.10: Layer-wise distribution of semantic concepts across six high-level categories proposed in [53] on CLIP ViT-B/16. Each curve shows the average proportion of concept words associated with a given category that are matched in each transformer layer, evaluated independently for each dataset.

# G  Understanding the Role of Intermediate Layers

**Empirical Validation of Entropy-based Selection.** Our entropy-based selection criterion is empirically validated in Figure G.11, which shows the relationship between entropy and the average false positive rate (FPR@95) across multiple challenging OOD datasets (iNaturalist, Textures, SUN, and Places) using CLIP ViT-B/32. Each point corresponds to a specific combination of intermediate layers.

Two distinct clusters emerge: low-entropy combinations achieving low FPR@95, and high-entropy combinations approaching baseline performance. A consistent positive correlation between entropy and FPR@95 is observed across architectures, suggesting that entropy can be exploited to unsupervisedly select promising layer combinations for OOD detection.

Importantly, we find that this relationship strengthens when restricting the maximum combination length. Shorter combinations yield more distinct clustering toward low entropy and low FPR@95, while longer combinations introduce redundancy and noise, weakening the discriminative signal.

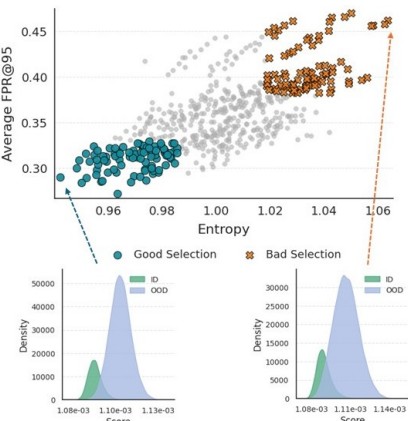

Figure G.11: Entropy vs. FPR@95 for different layer combinations, highlighting good and bad selections. Bottom: ID and OOD score distributions for representative examples.

**Effect of the size of the combinations in relation between average FPR and entropy.** We observe that constraining the maximum combination length strengthens the relationship between entropy and detection performance. Specifically, shorter combinations yield more coherent clustering toward low entropy and low FPR@95, as they are less likely to include noisy or redundant layers. In contrast, longer combinations often aggregate conflicting or less informative signals, diluting discriminative power and degrading OOD separation.

This trend suggests that limiting combination length acts as a natural form of regularization, reducing fusion noise and producing more robust, confidently separated layer subsets. These findings reinforce entropy as a principled selection criterion: when applied over constrained layer sets, it consistently identifies combinations that maximize zero-shot OOD detection performance. Further supporting evidence is provided in Section 6.

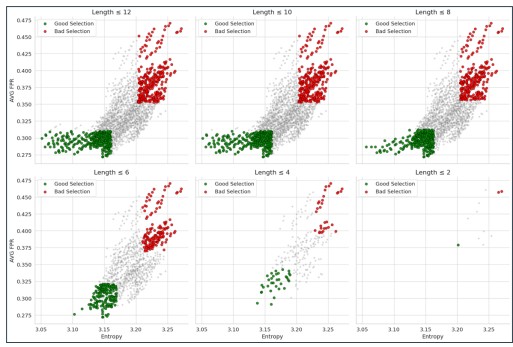
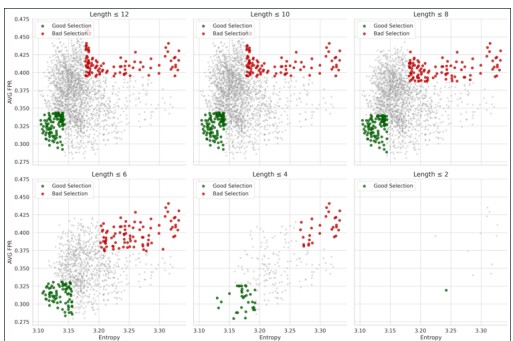

(a) ViT-B/32: Entropy vs. AVG FPR across varying combination lengths. Lower entropy correlates with improved OOD detection.

(b) ViT-B/16: Similar positive correlation between entropy and FPR, validating robustness of entropy-based selection.

Figure G.12: Empirical validation of entropy-based selection for OOD detection. Across both ViT-B/32 and ViT-B/16, lower entropy values correspond to lower false positive rates (FPR). Shorter combinations reduce noise, yielding more confident and effective selections.

# H  Why Intermediate Layers Help: A Theoretical View

**Confidence-based scoring across depth.** Let $x \in \mathcal{X}$ be an input sample and let $\phi_v^{(\ell)}(x) \in \mathbb{R}^{d_\ell}$ denote the visual feature at layer $\ell \in \mathcal{L}$ of the pretrained encoder. For pretrained classification models, we define $S_{n,\ell} = \max_{y \in \mathcal{Y}} \mathrm{Softmax}_c(h^{(\ell)}(\phi_v^{(\ell)}(x)))$ (MSP score). For contrastive vision-language models like CLIP, we follow the procedure in Section 2, applying the pretrained image-to-text projection $p$ to layer $\ell$, and define $S_{n,\ell} = \max_{y \in \mathcal{Y}} \mathrm{Softmax}_y(\cos(p(\phi_v^{(\ell)}(x)), T(y))/\tau)$ (MCM score). These scores form the confidence tensor $S \in \mathbb{R}^{N \times L}$, where each element $S_{n,\ell}$ is the scalar OOD score for instance $n$ at layer $\ell$.

**Motivation for fusion.** For a subset of layers $\mathcal{L} \subseteq \{1, \dots, L\}$, define the fused score as:

$$\bar{S}_n = \frac{1}{|\mathcal{L}|} \sum_{\ell \in \mathcal{L}} S_{n,\ell},$$

where $\bar{S}_n$ represents the average confidence score across selected layers for instance $n$ (e.g the set of layers $\{1, 2, 12\}$). Fusion strategies aim to reduce noise and increase robustness by integrating complementary representations across depth.

**Assumptions for effective fusion.** We propose the following sufficient conditions for the effectiveness of intermediate-layer fusion:

**A1  Semantic progression:** Deeper layers encode more task-aligned representations; shallow layers remain task-agnostic.

**A2  Representation diversity:** Layer-level scores $\{S_{n,\ell}\}_{\ell \in \mathcal{L}}$ are not perfectly correlated, i.e., $\mathrm{Var}[\bar{S}_n] > 0$.

**A3 Prediction consistency:** Scores fluctuate more for OOD than ID samples:

$$\text{Var}_{\text{ID}}[S_{n,\ell}] < \text{Var}_{\text{OOD}}[S_{n,\ell}].$$

**Theoretical insight.** Under mild assumptions—bounded scores, non-positive pairwise covariances (empirically supported in Fig. 3), and conditions **A1–A3**—we can show that fusion reduces variance in OOD scores and increases the separation between ID and OOD distributions in expectation.

**Proposition 1** (Fusion reduces variance and amplifies ID–OOD gap). *Let $\bar{S}(x)$ denote the average confidence score across a subset of layers $\mathcal{L} \subseteq \{1, \dots, L\}$. Then:*

$$\text{Var}_{OOD}[\bar{S}(x)] < \frac{1}{|\mathcal{L}|^2} \sum_{\ell \in \mathcal{L}} \sigma_\ell^2, \qquad \mathbb{E}_{X_{ID}}[\bar{S}(x)] - \mathbb{E}_{X_{OOD}}[\bar{S}(x)] > \max_{\ell \in \mathcal{L}} \left( \mathbb{E}_{X_{ID}}[S_{n,\ell}] - \mathbb{E}_{X_{OOD}}[S_{n,\ell}] \right).$$

*Thus, fusion reduces score variance on OOD data and increases the expected confidence gap between ID and OOD samples—yielding improved detection relative to any individual layer.*

> **Discussion**
>
> *The theoretical conditions outlined above are well-aligned with our empirical observations. CLIP models exhibit high inter-layer representational diversity (as indicated by low SVCCA similarity) and stable prediction behavior (as reflected by low top-1 disagreement), satisfying both A2 and A3. As shown in Figure 1, these properties translate into consistent gains from multi-layer fusion. In contrast, models such as MAE and DINOv2—characterized by limited entropy variation and unstable intermediate predictions—frequently violate these assumptions, resulting in diminished or even detrimental effects when layers are combined.*

# I  Random Projection for Resnet Architecture

To harmonize feature map dimensions across architectures and layers—and to enable alignment with a shared text embedding space—we apply a lightweight two-stage projection. Given a feature tensor $\mathbf{F} \in \mathbb{R}^{C_{\text{in}} \times H \times W}$, we first apply adaptive average pooling to resize the spatial dimensions to a fixed target shape $(H_{\text{target}}, W_{\text{target}})$:

$$\mathbf{F}_{\text{pooled}} = \text{AdaptiveAvgPool2D}(\mathbf{F}, (H_{\text{target}}, W_{\text{target}}))$$

Next, a $1 \times 1$ convolution projects the feature channels to $C_{\text{target}}$, which matches the dimensionality of the text features:

$$\mathbf{F}_{\text{proj}} = \text{Conv2D}_{1 \times 1}(\mathbf{F}_{\text{pooled}}, C_{\text{target}})$$

This operation ensures that all visual representations, irrespective of their originating layer or architecture, are transformed into a common shape and an embedding space compatible with the text modality. We interpret the $1 \times 1$ convolution as a random linear projection over the channel dimension. Unlike spatial convolutions, this operation applies an independent linear transformation at each spatial location, akin to a fully connected layer applied per pixel. When randomly initialized (e.g., using He or Xavier initialization), such projections approximately preserve the geometry of the input space with high probability, as guaranteed by the Johnson–Lindenstrauss lemma [11, 36, 45]. This allows the projected visual features to remain discriminative while enabling seamless fusion or comparison with the corresponding text representations.

# J  Different architectures and prompt-based methods

To test portability across backbones and prompting strategies (Table J), we find intermediate-layer fusion remains effective beyond the default setting: with MCM on RN50x4 it yields AUROC +0.85; FPR −3.55, while ViT-L/14 improves more modestly. Prompted ViT-B/16 also benefits, with NegLabel (AUROC +0.84; FPR −1.61) and CSP (AUROC +0.98; FPR −1.99). These trends mirror our layer-agreement analysis: heterogeneous ResNet stages let fusion reduce variance and suppress spurious peaks with MCM. Over this, this results highlight that the random projections I are able to improve this performance and specific in this network the intermediate layers is able to improve the results in every ood dataset.

Whereas ViT-L/14's higher inter-layer redundancy and sharper softmax (with $\tau = 0.01$) leave less headroom. Dataset context matters: the largest FPR drops appear on scene-centric SUN/Places where early and mid layers retain layout/background cues; gains are smaller on DTD and iNaturalist, where final embeddings already align with object-centric semantics. This difficult can be generated by the deepness of the network that generate a mordissimilar behavior acrees the extra layer that has in comparison with vit-B/16 ad vit=b/32 and the entropy in its output as we show in Sec 3.

Prompt-based methods such as CSP [7] and NegLabel [34] typically rely on external resources (e.g., WordNet hierarchies) or handcrafted prompts and use method-specific scoring. Our approach operates at the representation level by fusing intermediate layers, making it orthogonal to prompt design and broadly compatible with existing scoring rules. Applied to NegLabel, fusion yields consistent improvements across most datasets. CSP, although more memory-intensive, often benefits even more, likely because its synthetic negatives enlarge the ID vs OOD margin and better exploit complementary mid-layer cues.

Table 6: **Training-free OOD detection with additional backbones and prompt-based methods** (ImageNet-1K as ID). *Base*: without intermediate fusion; *+Fusion*: our training-free intermediate fusion. Best per column in **bold**.

| Method | Backbone | iNaturalist | | SUN | | Places | | Texture | | Average | |
|---|---|---|---|---|---|---|---|---|---|---|---|
| | | FPR↓ | AUC↑ | FPR↓ | AUC↑ | FPR↓ | AUC↑ | FPR↓ | AUC↑ | FPR↓ | AUC↑ |
| **Extra backbones (MCM)** | | | | | | | | | | | |
| MCM (Last Layer) | RN50x4 | 44.00 | 91.59 | 35.13 | 92.83 | 44.30 | 89.38 | 57.22 | 85.99 | 45.16 | 89.95 |
| MCM (Intermediate) | RN50x4 | **41.79** | **92.00** | **31.83** | **93.48** | **40.46** | **90.36** | **52.38** | **87.37** | **41.61** | **90.80** |
| MCM (Last Layer) | ViT-L/14 | **24.91** | 95.44 | 29.58 | 93.98 | 35.51 | 92.02 | **58.65** | **85.19** | **37.16** | 91.66 |
| MCM (Intermediate) | ViT-L/14 | 25.15 | **95.45** | **29.21** | **94.07** | **35.02** | **92.13** | 58.99 | 85.14 | 37.09 | **91.70** |
| **Prompt methods (composition with ViT-B/16)** | | | | | | | | | | | |
| NegLabel | ViT-B/16 | **1.91** | **99.49** | **20.53** | **95.49** | 35.59 | 91.64 | 43.56 | 90.22 | 25.40 | 94.21 |
| NegLabel + Int. Layers | ViT-B/16 | 2.35 | 99.36 | 21.10 | 95.38 | **32.31** | **93.35** | **39.38** | **92.10** | **23.79** | **95.05** |
| CSP | ViT-B/16 | 1.54 | 99.60 | 13.66 | 96.66 | 29.32 | 92.90 | 25.52 | 93.86 | 17.51 | 95.76 |
| CSP + Int. Layers | ViT-B/16 | **1.48** | **99.64** | **10.90** | **97.59** | **26.97** | **93.97** | **22.73** | **95.75** | **15.52** | **96.74** |

# K   Orthogonality to Scoring Rules

Consider logits $\mathbf{z} \in \mathbb{R}^K$, where $K$ denotes the number of classes. Following our experimental setup (Section 3), we analyze OOD detection methods: MaxLogit ($\max_i z_i$), MCM ($\max_i p_i$, with $p_i = \frac{\exp(z_i)}{\sum_j \exp(z_j)}$), Energy ($-\log \sum_{i=1}^{K} \exp(z_i)$) [44], Entropy ($-\sum_{i=1}^{K} p_i \log p_i$), and Variance ($\sum_{i=1}^{K} p_i(p_i - \bar{p})^2$).

The differential improvements observed across methods reflect their computational characteristics. MaxLogit [25] and MCM [47] rely directly on peak logit or probability values, making them particularly sensitive to intermediate layer representations, where the separation between ID and OOD peaks, $\Delta_{\text{peak}}^{(\ell)} = \mathbb{E}[\max_i z_i^{(\ell)}|\text{ID}] - \mathbb{E}[\max_i z_i^{(\ell)}|\text{OOD}]$, is typically larger due to reduced saturation effects. The observed improvement for MCM specifically benefits from the softmax transformation applied over logits, which amplifies probability differences and enhances discriminative power between ID and OOD samples. This theoretical insight is empirically validated (Table K), showing substantial intermediate-layer improvements for MCM (average $+10.2\%$, notably $+16.6\%$ for ViT-B/32) and consistent gains for MaxLogit ($+4.3\%$).

In contrast, Energy [44], Entropy, and Variance methods aggregate global distributional information through log-sum-exp operations and probability averaging, making them inherently less sensitive to peak-specific variations and thus displaying more modest improvements ($+0.4\%$, $+6.2\%$, and $+6.1\%$, respectively). Architectural differences further modulate these improvements. Our analysis reveals that CLIP models demonstrate broad prediction agreement across layers (Figure 4), promoting stable predictions conducive to effective fusion, whereas supervised models show independent prediction evolution across layers. These architectural patterns, combined with the representational diversity captured in our entropy calibration analysis (Figure M.18), explain the variance in observed improvements, with ViT-B/32 achieving the highest gains due to optimal balance between inter-layer diversity and prediction consistency.

Table 7: **Comprehensive OOD detection results using intermediate vs last layer features.** We report the false positive rate at 95% TPR (FPR95, ↓) and AUROC (↑) across four OOD datasets: SUN, Places365, DTD, and iNaturalist. Each method is evaluated using ResNet, ViT-B/32, or ViT-B/16 backbones with both last and intermediate layer features. Best results per backbone are highlighted in **bold**.

| Method | Layer | Backbone | SUN | | Places365 | | DTD | | iNaturalist | | Average | |
|---|---|---|---|---|---|---|---|---|---|---|---|---|
| | | | FPR↓ | AUC↑ | FPR↓ | AUC↑ | FPR↓ | AUC↑ | FPR↓ | AUC↑ | FPR↓ | AUC↑ |
| **ResNet Results** | | | | | | | | | | | | |
| Entropy | Last | ResNet | 55.84 | 88.04 | 72.42 | 79.31 | 73.32 | 78.78 | 83.13 | 71.80 | 71.18 | 79.48 |
| Entropy | Intermediate | ResNet | 55.69 | 88.06 | 72.91 | 79.14 | 73.81 | 78.74 | 82.19 | 72.46 | 71.15 | 79.60 |
| Energy | Last | ResNet | 98.63 | 39.20 | 94.56 | 52.88 | 99.70 | 20.35 | 98.83 | 43.01 | 97.93 | 38.86 |
| Energy | Intermediate | ResNet | 98.49 | 39.65 | 94.12 | 53.43 | 99.68 | 19.79 | 98.57 | 44.53 | 97.72 | 39.35 |
| Variance | Last | ResNet | 55.09 | 88.19 | 72.03 | 79.51 | 73.14 | 78.84 | 82.90 | 72.15 | 70.79 | 79.67 |
| Variance | Intermediate | ResNet | 55.01 | 88.22 | 72.49 | 79.35 | 73.53 | 78.80 | 81.88 | 72.80 | 70.73 | 79.79 |
| MaxLogit | Last | ResNet | 59.73 | 89.11 | 52.05 | 89.19 | 90.05 | 71.19 | 65.44 | 88.23 | 66.82 | 84.43 |
| MaxLogit | Intermediate | ResNet | 54.71 | 90.08 | 46.78 | 90.35 | 88.76 | 72.89 | 61.00 | 89.19 | 62.81 | 85.63 |
| MCM | Last | ResNet | 35.13 | 92.83 | 44.30 | 89.38 | 57.22 | 85.99 | 44.00 | 91.59 | 45.16 | 89.95 |
| MCM | Intermediate | ResNet | **31.83** | **93.48** | **40.46** | **90.36** | **52.38** | **87.37** | **41.79** | **92.00** | **41.61** | **90.80** |
| **ViT-B/32 Results** | | | | | | | | | | | | |
| Entropy | Last | ViT-B/32 | 59.13 | 84.38 | 67.38 | 79.81 | 76.72 | 72.08 | 73.34 | 79.29 | 69.14 | 78.89 |
| Entropy | Intermediate | ViT-B/32 | 65.09 | 80.03 | 84.96 | 68.69 | 66.86 | 80.54 | 49.72 | 90.68 | 66.66 | 79.98 |
| Energy | Last | ViT-B/32 | 97.77 | 42.25 | 95.79 | 53.02 | 98.92 | 26.50 | 99.79 | 37.98 | 98.07 | 39.94 |
| Energy | Intermediate | ViT-B/32 | 97.64 | 45.85 | 96.07 | 57.90 | 97.29 | 40.18 | 99.88 | 32.71 | 97.72 | 44.16 |
| Variance | Last | ViT-B/32 | 58.78 | 84.60 | 67.10 | 80.06 | 76.63 | 72.18 | 73.11 | 79.58 | 68.91 | 79.11 |
| Variance | Intermediate | ViT-B/32 | 64.92 | 80.21 | 84.84 | 68.94 | 66.91 | 80.64 | 49.67 | 90.78 | 66.59 | 80.14 |
| MaxLogit | Last | ViT-B/32 | 64.98 | 86.78 | 56.36 | 88.02 | 87.00 | 70.45 | 65.22 | 87.58 | 68.39 | 83.21 |
| MaxLogit | Intermediate | ViT-B/32 | 69.30 | 83.70 | 67.89 | 83.95 | 80.25 | 72.03 | 54.82 | 89.56 | 68.06 | 82.31 |
| MCM | Last | ViT-B/32 | 40.99 | 91.56 | 46.71 | 89.25 | 60.90 | 85.03 | 33.85 | 93.62 | 45.61 | 89.87 |
| MCM | Intermediate | ViT-B/32 | **28.98** | **93.37** | **35.69** | **91.61** | **39.36** | **91.07** | **12.02** | **97.64** | **29.01** | **93.42** |
| **ViT-B/16 Results** | | | | | | | | | | | | |
| Entropy | Last | ViT-B/16 | 68.00 | 81.36 | 74.90 | 76.35 | 79.08 | 72.37 | 87.00 | 65.05 | 77.24 | 73.78 |
| Entropy | Intermediate | ViT-B/16 | 66.86 | 78.25 | 81.86 | 72.64 | 36.33 | 92.12 | 59.78 | 88.25 | 61.21 | 82.81 |
| Energy | Last | ViT-B/16 | 98.68 | 37.66 | 97.21 | 45.39 | 99.22 | 24.68 | 99.77 | 36.95 | 98.72 | 36.17 |
| Energy | Intermediate | ViT-B/16 | 98.49 | 41.02 | 99.48 | 33.19 | 94.52 | 38.66 | 99.87 | 41.52 | 98.09 | 38.60 |
| Variance | Last | ViT-B/16 | 67.54 | 81.61 | 74.47 | 76.62 | 78.97 | 72.47 | 86.67 | 65.62 | 76.91 | 74.08 |
| Variance | Intermediate | ViT-B/16 | 66.73 | 78.40 | 81.51 | 72.98 | 36.21 | 92.13 | 59.21 | 88.45 | 60.91 | 82.99 |
| MaxLogit | Last | ViT-B/16 | 64.45 | 87.43 | 60.40 | 87.07 | 85.48 | 72.56 | 60.56 | 89.64 | 67.72 | 84.17 |
| MaxLogit | Intermediate | ViT-B/16 | 70.32 | 85.04 | 69.05 | 84.15 | 50.51 | 88.18 | 46.61 | 91.97 | 59.12 | 87.33 |
| MCM | Last | ViT-B/16 | 37.41 | 92.57 | 43.67 | 89.96 | 57.34 | 86.18 | 30.67 | 94.63 | 42.27 | 90.83 |
| MCM | Intermediate | ViT-B/16 | 45.58 | 89.69 | **35.71** | **92.72** | **25.51** | **94.84** | **15.98** | **96.90** | **30.70** | **93.54** |

Formally, for any such score ($s$), an increase in $\Delta_s := \mathbb{E}\left[s\left(z^{\text{fused}}\right) \mid \text{ID}\right] - \mathbb{E}\left[s\left(z^{\text{fused}}\right) \mid \text{OOD}\right]$ improves thresholded decisions for that ($s$), indicating robustness that is orthogonal to the choice of scoring rule.

### Discussion

**Fusing intermediate representations offers a training-free, plug-and-play mechanism that consistently strengthens OOD robustness across architectures, datasets, and prompting regimes while adding only modest compute and memory cost.** The effect arises from **complementary early- and mid-layer signals that widen the separation between familiar and unfamiliar inputs and stabilize predictions**. Because the mechanism acts at the representation level, it integrates without changing existing pipelines. Gains can vary with architectural traits; models whose information is concentrated in late layers may show smaller improvements. Future directions include adaptive per-model layer selection, dynamic fusion policies at inference time, and exploring token- or region-level cues to further enhance reliability in open-world settings.

# L    Computational Cost Analysis of Intermediate Layer Usage

We evaluate the computational implications of incorporating intermediate layer features across three CLIP architectures: ResNet-50x4, ViT-B/16, and ViT-B/32. The analysis spans batch sizes from 1 to 8 and considers peak GPU memory, per-image latency, total inference time, and throughput (images processed per second). As shown in Figures L.13, L.15, and L.14, the additional operations required for intermediate feature extraction introduce limited overhead. Across all configurations, memory and latency values remain within practical deployment limits, supporting the method's suitability for real-world use.

ViT-B/32 exhibits the most efficient resource profile (Figure L.14), with memory overhead ranging from $1.7\%$ to $7.1\%$ (average: $3.4\%$) across batch sizes and average latency changes of just $3.3\%$. Notably, ViT-B/32 demonstrates latency reductions at batch sizes 4 and 8 ($-1.1\%$ and $-1.0\%$, respectively), indicating improved hardware utilization. ViT-B/16 shows higher memory overhead (average: $10.2\%$, up to $20.6\%$ at batch size 8), yet achieves an overall latency *reduction* of $5.7\%$ on average, with throughput impact of only $-0.7\%$ (Figure L.15). Both ViT architectures exhibit counterintuitive latency improvements at larger batch sizes, suggesting better parallelization and GPU kernel efficiency when processing intermediate features. ResNet-50x4 exhibits more conventional scaling behavior (Figure L.13), with a stable memory overhead averaging $12.4\%$ and latency increases ranging from $5.1\%$ to $14.7\%$ (mean $11.1\%$). Throughput decreases by an average of $10.7\%$, primarily due to the use of external random projections to align the dimensionality of intermediate-layer features.

Importantly, absolute performance values remain deployment-friendly in all cases. Peak memory usage stays under 0.7 GB even at batch size 8, and per-image latency remains compatible with both real-time and high-throughput applications (maintaining 8-60 images/second depending on architecture and batch size). These results confirm that the use of intermediate features is computationally viable across architectures, offering substantial detection performance improvements-up on far-OOD benchmarks (see Tables 5.1 and 2) and consistent gains across diverse scoring functions (Table K) with minimal computational cost. Full breakdowns of the quantitative results are reported in Tables L, L, and L.

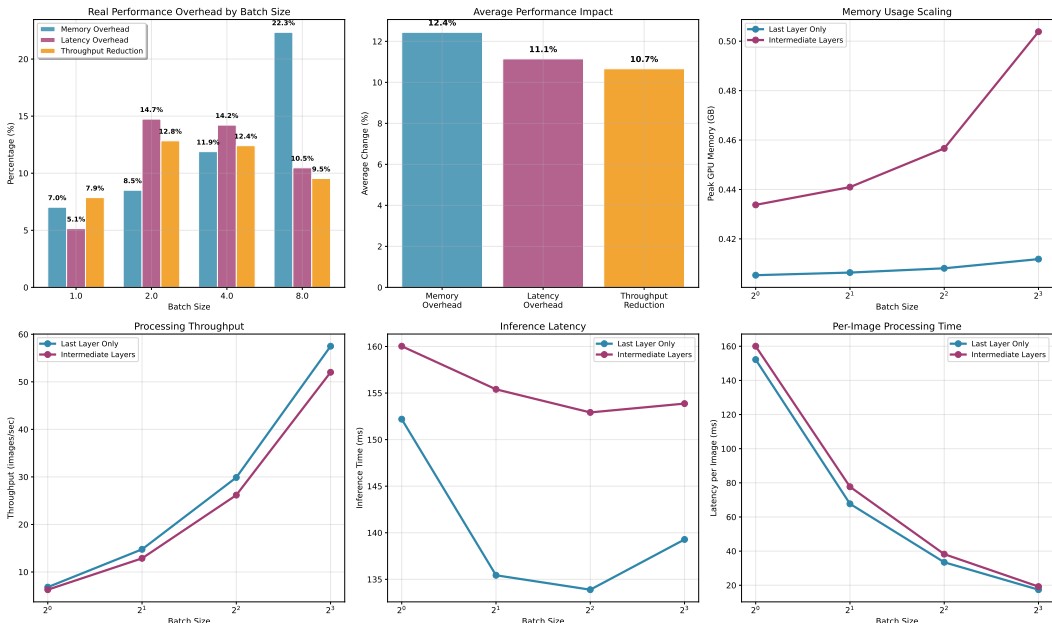

Figure L.13: Computational cost comparison for **CLIP RN50x4** using intermediate layers versus last-layer only inference.

Table 8: **ResNet-50x4 Performance Analysis: Last Layer vs All Layers.** We report memory usage (GB), latency per image (ms), and overhead percentages (%, ↑ indicates increase) for CLIP ResNet-50x4 architecture. Results show mean ± standard deviation across 5 measurement runs per configuration.

| Batch Size | Memory Usage (GB) | | Latency per Image (ms) | | Overhead (%) | | Runs (L/A) |
|---|---|---|---|---|---|---|---|
| | Last Layer | All Layers | Last Layer | All Layers | Memory↑ | Latency↑ | |
| 1 | 0.41±0.00 | 0.43±0.00 | 152.20±33.17 | 160.03±11.58 | +7.0 | +5.1 | 5/5 |
| 2 | 0.41±0.00 | 0.44±0.00 | 67.72±0.56 | 77.70±1.01 | +8.5 | +14.7 | 5/5 |
| 4 | 0.41±0.00 | 0.46±0.00 | 33.47±0.36 | 38.23±0.86 | +11.9 | +14.2 | 5/5 |
| 8 | 0.41±0.00 | 0.50±0.00 | 17.41±0.54 | 19.23±0.33 | +22.3 | +10.5 | 5/5 |

**Average Overhead:** Memory +12.4%, Latency +11.1%

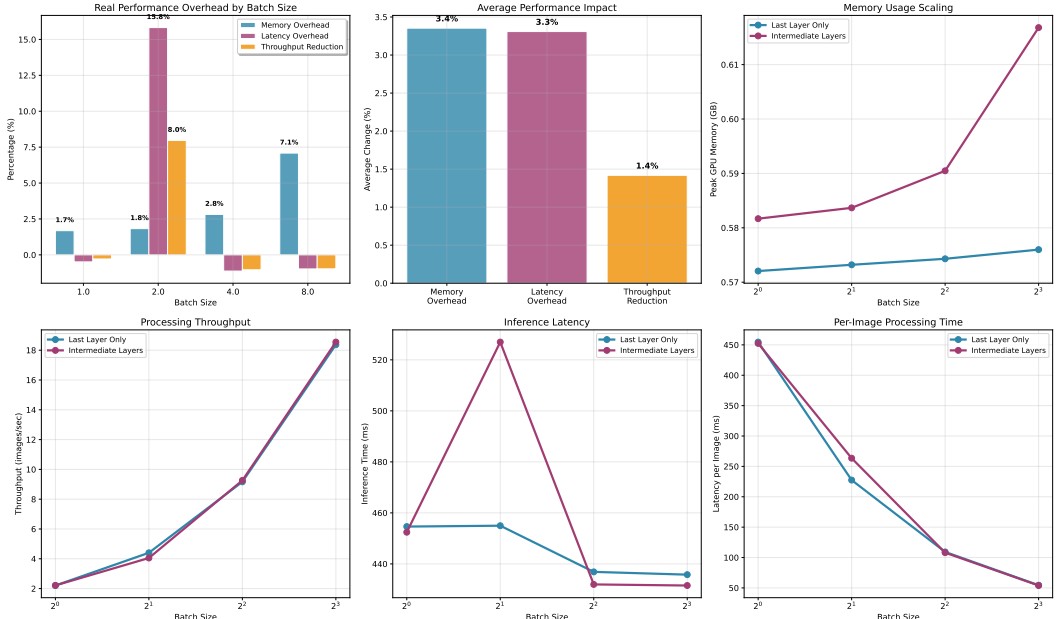

Figure L.14: Computational cost comparison for **CLIP ViT-B/32** using intermediate layers versus last-layer only inference.

Table 9: **ViT-B/32 Performance Analysis: Last Layer vs All Layers.** We report memory usage (GB), latency per image (ms), and overhead percentages (%, ↑ indicates increase, ↓ indicates decrease) for CLIP ViT-B/32 architecture. Results show mean ± standard deviation across 5 measurement runs per configuration.

| Batch Size | Memory Usage (GB) | | Latency per Image (ms) | | Overhead (%) | | Runs (L/A) |
|---|---|---|---|---|---|---|---|
| | Last Layer | All Layers | Last Layer | All Layers | Memory↑ | Latency↑ | |
| 1 | 0.57±0.00 | 0.58±0.00 | 454.64±25.75 | 452.42±14.36 | +1.7 | -0.5 | 5/5 |
| 2 | 0.57±0.00 | 0.58±0.00 | 227.49±9.11 | 263.49±79.81 | +1.8 | +15.8 | 5/5 |
| 4 | 0.57±0.00 | 0.59±0.00 | 109.22±3.74 | 107.99±1.16 | +2.8 | -1.1 | 5/5 |
| 8 | 0.58±0.00 | 0.62±0.00 | 54.47±1.30 | 53.94±1.11 | +7.1 | -1.0 | 5/5 |

**Average Overhead:** Memory +3.4%, Latency +3.3%

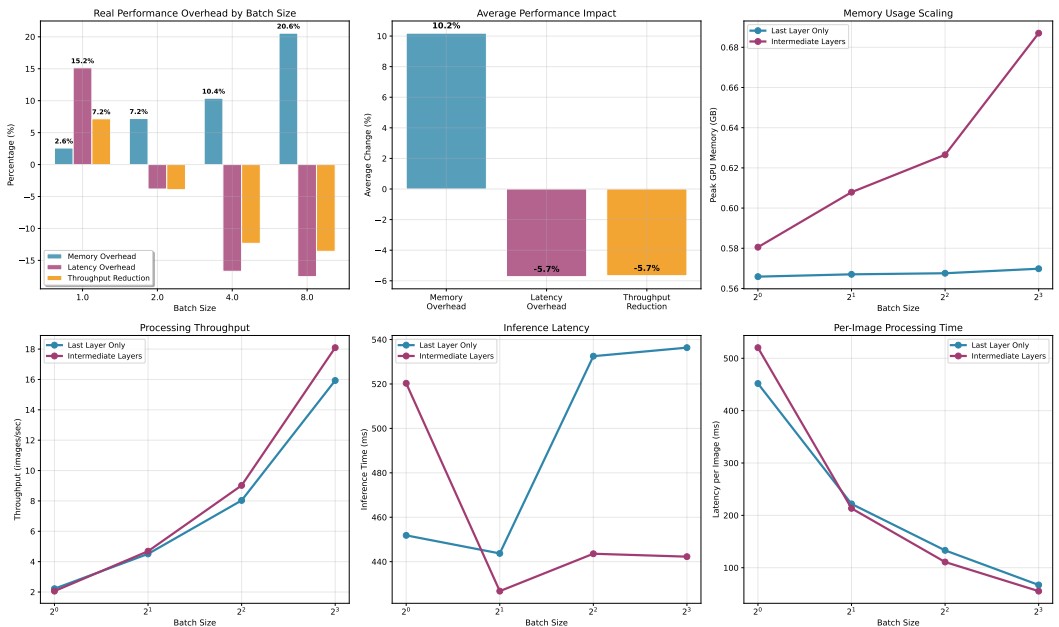

Figure L.15: Computational cost comparison for **CLIP ViT-B/16** using intermediate layers versus last-layer only inference.

Table 10: **ViT-B/16 Performance Analysis: Last Layer vs All Layers.** We report memory usage (GB), latency per image (ms), and overhead percentages (%, ↑ indicates increase, ↓ indicates decrease) for CLIP ViT-B/16 architecture. Results show mean ± standard deviation across 5 measurement runs per configuration.

| Batch | Memory Usage (GB) | | Latency per Image (ms) | | Overhead (%) | | Runs |
|---|---|---|---|---|---|---|---|
| **Size** | Last Layer | All Layers | Last Layer | All Layers | Memory↑ | Latency↑ | (L/A) |
| 1 | 0.57±0.00 | 0.58±0.00 | 451.84±20.45 | 520.31±162.03 | +2.6 | +15.2 | 5/5 |
| 2 | 0.57±0.00 | 0.61±0.00 | 221.86±5.01 | 213.37±1.67 | +7.2 | -3.8 | 5/5 |
| 4 | 0.57±0.00 | 0.63±0.00 | 133.13±39.96 | 110.89±2.09 | +10.4 | -16.7 | 5/5 |
| 8 | 0.57±0.00 | 0.69±0.00 | 67.04±20.34 | 55.28±0.94 | +20.6 | -17.5 | 5/5 |

**Average Overhead:** Memory +10.2%, Latency -5.7%

# M   Additional Figures

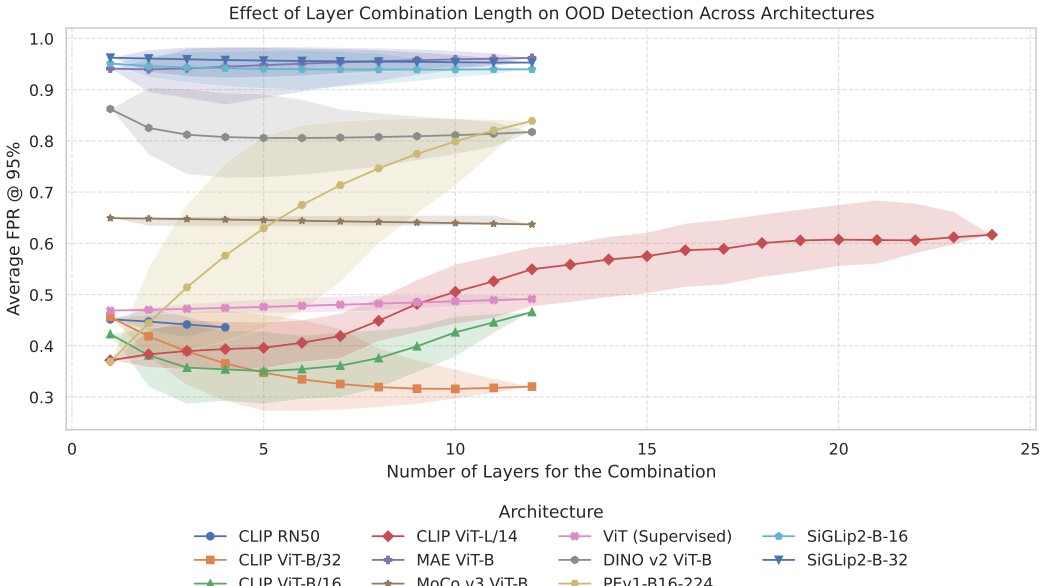

Figure M.16: Impact of combination length on OOD detection performance across architectures. The plot reports average FPR@95% as a function of the number of fused layers. Combining a moderate number of layers typically improves performance, while longer combinations may degrade it.

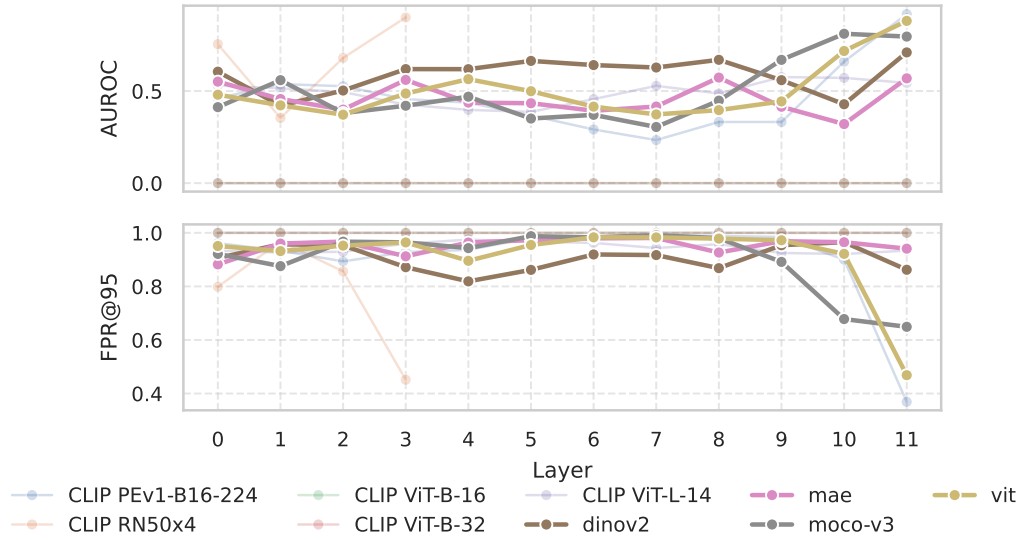

Figure M.17: Layer-wise OOD detection performance across architectures. Most architectures exhibit their best performance near the final layer, while early layers generally under-perform.

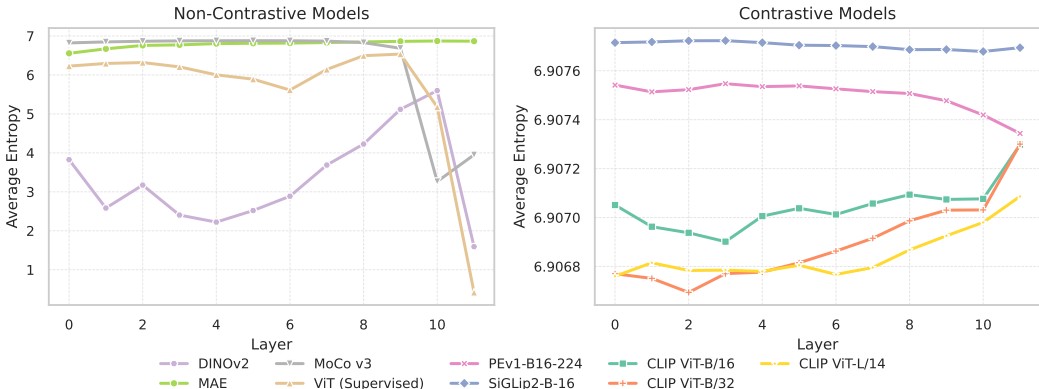

Figure M.18: **Average entropy across transformer layers for various vision models.** We compare softmax entropy across layers. Supervised models exhibit low entropy in later layers, reflecting overconfident predictions, while self-supervised and contrastive models (e.g., MAE, CLIP) show greater entropy variation across depth. The inset highlights subtle differences in the high-entropy regime across CLIP variants, SiGLip2, and PE.

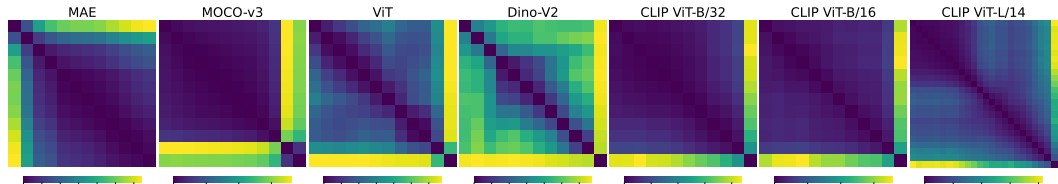

Figure M.19: Layer-wise Jensen-Shannon Divergence (JSD) between predicted class probabilities across different architectures. Each heatmap corresponds to a specific model and illustrates the pairwise JSD between layers. Color intensity reflects the degree of divergence, with brighter values indicating greater dissimilarity in output distributions. **Note that each subplot uses its own color scale to emphasize internal variation within each architecture**.

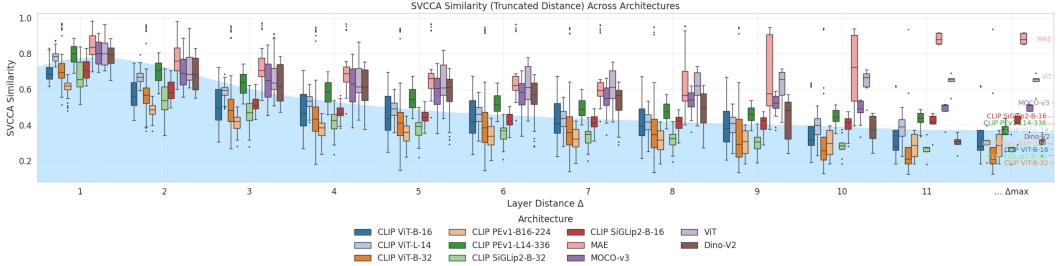

Figure M.20: SVCCA similarity vs. layer distance $\Delta$ across ViT architectures. Box plots show SVCCA distributions per $\Delta$; overlaid lines denote mean similarity.

