# OpenReview forum: "Mysteries of the Deep: Role of Intermediate Representations in Out of Distribution Detection"
_NeurIPS.cc/2025/Conference — NeurIPS 2025 poster_

### Official Review · Reviewer_V3xf · 2025-06-25

**Clarity:** 4
**Significance:** 3
**Originality:** 2
**Rating:** 4
**Confidence:** 4

**Summary:**

The paper studies the how OOD detection changes when information from the intermediate layers of models is included, rather than focusing on the final layer representations. The paper evaluates the OOD performance across many architectures and benchmarks, and also presents a layer selection method which increases OOD detection performance.

**Questions:**

- Which layers get chosen from the different layer selection methods? For example, for the entropy-based method, do models typically get more confident as we progress through each layer, meaning that optimal combination could just be the last n layers? It would also be helpful to incorporate a few other baselines such as "single best layer" or "penultimate layer" in Figure 6 so I have a better sense of how each method compares.
- We see in Figure 1 that increases the number of layers significantly hurts performance for PEv1-B16-224. Do you have any hypotheses for why this is happening? Do the layer selection methods choose N=1 for this model, or would they suggest larger Ns which would hurt performance?
- In *Surgical Fine-Tuning Improves Adaption to Distribution Shifts* (Lee et al), the authors note the earlier layers seem to be helpful for input-level shifts, middle layers for feature-level shifts, etc for OOD generalization. Do you notice similar trends (or other trends) with your method for OOD detection, and do you have any intuitions for what layers are typically helpful for specific types of OOD detection problems?

**Ethical Concerns:**

["NO or VERY MINOR ethics concerns only"]

**Final Justification:**

The authors explore an important aspect of OOD detection which often gets overlooked, and determine that features from middle to deep layers are often beneficial. This is an interesting finding, and their method is fairly straightforward and leads to positive performance. They also included more ablations during the rebuttal period which answered my questions about the types of features selected. However, there have been many previous works which explore the effectiveness of intermediate layers, so there is limited novelty. The entropy-based selection strategy could also have been better motivated.

**Limitations:**

Yes

**Quality:**

3

**Strengths And Weaknesses:**

Strengths
- The paper explores an important aspect of OOD detection, feature selection, where last-layer features are often used by default. Incorporating intermediate layers seems to generally have a positive impact on performance, and the authors also demonstrated results over a wide variety of settings and models.
- I appreciate the number of ablations in the paper, and they really help illuminate the difference in behavior between model architectures. For example, I thought Figure 3 was really interesting and highlighted the differences in behavior between contrastive vs supervised models. The authors did a thorough job of benchmarking the impacts of many components of their method, such as the number of layers, layer selection method, and other hyperparameters.
- The paper is well-written, and the motivation for each section of the paper is clear.

Weaknesses
- There are quite a few papers which explore the effectiveness of intermediate layers for OOD detection. Although many of these works do not focus on vision-language models, many of them come to the same conclusions that we should consider more than just the penultimate layer.
  1. Detecting Out-of-Distribution Inputs in Deep Neural Networks Using an Early-Layer Output (Abdelzad et al, 2019)
  2. MOOD: Multi-level Out-of-distribution Detection (Lin et al, 2021)
  3. Layer Adaptive Deep Neural Networks for Out-of-Distribution Detection (Wang et al, 2022)
  4. Multi-layer Aggregation as a Key to Feature-Based OOD Detection (Lambert et al, 2023)
- The proposed entropy minimization layer selection method seems a bit unintuitive to me, since it is unclear why choosing the subset of features that the model is the most confident on for ID data would translate to improved ID vs OOD separation.
- The ablation of layer selection strategies could also be expanded on. It may be helpful to add a section in the appendix which clearly defines each of the methods and compare the differences in the layers which end up being selected.

---

> ### Author Rebuttal · Authors · 2025-07-30
>
> We appreciate the reviewer’s positive remarks on the paper’s clarity, experimental rigor, and architectural insights. Motivated by the suggestions, we have expanded our analysis in the revision to better characterize selection behaviors across metrics and architectures, and to clarify the benefits and limitations of our entropy-based strategy.
>
> ### **Q1: Which Layers Are Chosen by Each Selection Method?**
>
> We thank the reviewer for the question. To clarify the behavior of each selection method, we will include a dedicated section in the appendix describing the layers typically selected by different metrics. Below, we provide representative examples:
>
> | **ENTROPY**                    | **KURTOSIS**             | **GINI**              | **JSD**                    |
> | ------------------------------ | ------------------------ | --------------------- | -------------------------- |
> | \[3, 4, 6, 8, 11] — FPR: 0.307 | \[0, 11] — FPR: 0.435    | \[3, 11] — FPR: 0.408 | \[0,1,2,9,11] — FPR: 0.367 |
> | \[2, 3, 6, 8, 11] — FPR: 0.298 | \[11] — FPR: 0.423       | \[5, 11] — FPR: 0.411 | \[0,1,3,9,11] — FPR: 0.381 |
> | \[3, 4, 7, 8, 11] — FPR: 0.315 | \[0, 1, 11] — FPR: 0.441 | \[1, 11] — FPR: 0.429 | \[0,2,3,9,11] — FPR: 0.364 |
> | \[2, 3, 4, 8, 11] — FPR: 0.316 | \[1, 11] — FPR: 0.429    | \[0, 11] — FPR: 0.435 | \[0,1,2,7,11] — FPR: 0.342 |
>
> These results show that **entropy consistently selects a mix of mid-to-deep layers** (e.g., 3–8, 11), rather than simply favoring the final layers. In contrast, kurtosis, Gini, and JSD often focus on shallow–final pairs or layer 11 alone, leading to reduced performance. A qualitative explanation is provided in Figures G.11–G.12, where **only entropy exhibits a stable correlation with FPR\@95**.
>
> ### **Q2: Do Certain Layers Consistently Contribute More to Performance?**
>
> To support the intuition that some layers are more valuable than others, we conducted a **marginal utility analysis** using ViT-B/16 on ImageNet. We ranked layers by their individual contribution to FPR reduction and AUROC improvement:
>
> | Rank | Layer | FPR Reduction (%) | AUC Improvement (%) | Overall Score |
> | ---- | ----- | ----------------- | ------------------- | ------------- |
> | 1    | 8     | **+3.42**         | **+0.54**           | **1.98**      |
> | 2    | 6     | +2.93             | +0.69               |  1.81         |
> | 3    | 7     | +2.53             | +0.54               | 1.54          |
> | 4    | 4     | +1.65             | +0.40               | 1.03          |
> | 5    | 2     | +0.59             | +0.12               | 0.36          |
> | 6    | 9     | –0.14             | –0.32               | –0.23         |
> | 7    | 3     | –0.71             | –0.05               | –0.38         |
> | 8    | 1     | –0.83             | –0.19               | –0.51         |
> | 9    | 0     | –0.85             | –0.21               | –0.53         |
> | 10   | 5     | –2.36             | –0.64               | –1.50         |
> | 11   | 10    | –2.15             | –1.15               | –1.65         |
>
> This confirms that **layers 6–8 consistently provide strong utility** and complement the final layer, reinforcing the idea that **optimal combinations involve diverse mid-to-deep layers**, not just the last *n*.
>
> In addition, our **consistency and diversity analysis** (Figures E.8–E.9) shows:
>
> * **Within each ID dataset**, top-10 oracle combinations have low pairwise Jaccard distance (≈ 0.48), indicating stable and convergent selections.
> * **Across datasets**, the overlap drops (≈ 30–50%), and fixed layer sets perform worse (e.g., FPR: 0.300 vs 0.288 with entropy-based selection), emphasizing the benefit of **dataset-specific layer selection**.
>
> ### **Q3: Why Does Performance Degrade with More Layers in PEv1-B/16?**
>
> We hypothesize that the performance degradation in **PEv1-B/16-224** when increasing the number of combined layers is due to its **distilled architecture**. Specifically, PE B/16 is distilled from the larger PE G model \[1], which likely **compresses most of the useful information into the final layers**. This contrasts with CLIP models, which are trained end-to-end with contrastive objectives that promote **semantic diversity across all layers**.
>
> As a result, intermediate representations in PEv1-B/16 tend to be less informative or redundant, and **combining multiple layers introduces noise rather than complementary signals**, ultimately harming OOD performance.
>
> ### **Q4: Do Selection Methods Adapt to PEv1-B/16 Behavior?**
>
> Yes. Our entropy-based selection method adapts accordingly and does **not favor large N** for PEv1-B/16. For example, it selects a small combination $[0, 1, 11]$, reflecting an attempt to minimize degradation. However, even this small combination performs **slightly worse** than using the final layer alone (FPR = 0.3816 vs. 0.3792), indicating that **intermediate layers are not reliably useful** in this model.
>
> This outcome supports the idea that **layer fusion is less beneficial for distilled architectures**, where semantic abstraction is concentrated near the output and intermediate layers are poorly calibrated as shown in our analysis.
>
> [1] Bolya et al. *Perception Encoder: The best visual embeddings are not at the output of the network*

---

> > ### Comment · Reviewer_V3xf · 2025-08-05
> >
> > Thanks for the responses! The new analyses regarding the importance of the mid to deep layers of the network are interesting. I have decided to retain my score of "borderline accept".

---

### Official Review · Reviewer_A4Qo · 2025-06-27

**Clarity:** 3
**Significance:** 2
**Originality:** 2
**Rating:** 4
**Confidence:** 4

**Summary:**

This paper proposes to leverage the intermediate layers of pre-trained models can encode surprisingly rich and diverse signals for detecting distributional shifts. To exploit latent representation diversity across layers, this paper introduces an entropybased criterion to automatically identify layers offering the most complementary information in a training-free setting—without access to OOD data. Estensive experiments across seven vision backbones have been conducted to evaluate the propsoed method.

**Questions:**

See Weaknesses.

**Ethical Concerns:**

["NO or VERY MINOR ethics concerns only"]

**Final Justification:**

The authors’ response effectively addressed my concerns, so I have decided to raise my score to "borderline accept".

**Limitations:**

The idea of leveraging intermediate layers to enhance OOD detection has been proposed in several prior works and the reported results are not state-of-the-art.  More details can be seen in Weaknesses.

**Quality:**

3

**Strengths And Weaknesses:**

**Strengths:**
1. The paper is well-structured and easy to follow.
2. It has been experimentally validated on various pre-trained models and is supported by a comprehensive ablation study.

**Weaknesses:**

1. Limited Novelty: I have to say, the idea of leveraging intermediate layers to enhance OOD detection has been proposed in several prior works [1,2,3]. However, the authors have not discussed these related methods. Please elaborate on how the proposed approach differs from and improves upon these existing works.

    [1] Lin, Ziqian, Sreya Dutta Roy, and Yixuan Li. "Mood: Multi-level out-of-distribution detection." Proceedings of the IEEE/CVF conference on Computer Vision and Pattern Recognition. 2021.

    [2] Guglielmo, Gianluca, and Marc Masana. "Leveraging Intermediate Representations for Better Out-of-Distribution Detection." arXiv preprint arXiv:2502.12849 (2025).

    [3] Fayyad, Jamil, et al. "Exploiting classifier inter-level features for efficient out-of-distribution detection." Image and Vision Computing 142 (2024): 104897.

2. Limited Scope of Scoring Methods: In Table 1 (based on CLIP ViT-B/16), why the discussion is limited to MCM scoring, do the conclusions generalize to other scoring strategies, such as energy score, MSP, and maxlogits? Please consider including results or discussion on these alternative scoring methods.

3. In Line 102, the paper mentions that a fixed random linear projection is used to address layer-wise dimensionality mismatches. The authors further claim in Appendix I that “such projections approximately preserve the geometry of the input space.” Could the authors provide experimental evidence to support this claim?

4. Does the use of intermediate layer features introduce additional GPU memory consumption? Please provide a comparison of the computational cost (e.g., memory usage, inference time) of the proposed method versus existing baselines.

5. The reported results are not state-of-the-art. The previous method, such as CSP [4],NegLabel [5],  Clipscope[6], have achieved much better performance comapred to the your results in Table 1.

    [4] Chen, Mengyuan, Junyu Gao, and Changsheng Xu. "Conjugated semantic pool improves ood detection with pre-trained vision-language models." Advances in Neural Information Processing Systems 37 (2024): 82560-82593.

    [5] Jiang, Xue, et al. "Negative label guided ood detection with pretrained vision-language models." arXiv preprint arXiv:2403.20078 (2024).

    [6] Fu, Hao, et al. "Clipscope: Enhancing zero-shot ood detection with bayesian scoring." arXiv preprint arXiv:2405.14737 (2024).

6. In Figure 6, why the layer selection strategies based on kurtosis, standard deviation, Gini coefficient, Jensen–Shannon (JSD) divergence peform even worse than random selection. Could the authors provide insights to explain this phenomenon?

---

> ### Author Rebuttal · Authors · 2025-07-30
>
> We thank the reviewer for their feedback. We have carefully considered each suggestion and incorporated new analyses, comparisons, and clarifications to address the concerns. We hope these additions better convey the motivation, novelty, and practical value of our work.
>
> ### **Q1: Relation to Prior Work Using Intermediate Representations**
>
> While prior work [1–3] has explored intermediate layers for OOD detection, our method differs in both scope and design. Previous approaches focus on supervised models, require labeled ID data, auxiliar networks and select a single best layer using trained classifiers or regularization. In contrast, our method is fully training-free, uses no ID labels, and operates directly on pretrained CLIP models without fine-tuning. We fuse multiple intermediate layers instead of selecting just one, capturing complementary OOD signals and achieving consistent gains across datasets and architectures. Unlike prior work limited to CNNs, we conduct a systematic analysis across seven models with diverse training objectives, including contrastive and self-supervised vision-language models like CLIP, SiGLip-v2, and Perception Encoder. Our study reveals new insights into the role of intermediate representations for OOD detection and is orthogonal to scoring functions, making it complementary to existing methods (see Questions 2 and 5).
>
>
> ### **Q2: Generalization to Other Scoring Methods**
>
> We thank the reviewer for this suggestion. Based on it, **we conducted additional evaluations using alternative scoring functions**. While our main focus is on **MCM** due to its strong and stable performance, our method is **compatible with a wide range of scoring strategies**. We will include the detailed results in the revised version.
>
>
> **ResNet (RN50x4)**
>
> | Method | Last Layer (FPR / AUROC) | Intermediate Fusion (FPR / AUROC) |
> |---|---|---|
> | MCM | 45.2 / 89.9 | 41.6 / 90.8 |
> | MaxLogit | 66.8 / 84.4 | 62.8 / 85.6 |
> | Entropy | 71.2 / 79.5 | 71.1 / 79.6 |
> | Variance | 70.8 / 79.7 | 70.7 / 79.8 |
> | Energy | 97.9 / 38.9 | 97.7 / 39.4 |
>
> **ViT-B/32**
>
> | Method | Last Layer (FPR / AUROC) | Intermediate Fusion (FPR / AUROC) |
> |---|---|---|
> | MCM | 45.6 / 89.8 | 29.0 / 93.4 |
> | MaxLogit | 68.4 / 83.2 | 68.0 / 82.3 |
> | Entropy | 69.1 / 78.9 | 66.7 / 79.9 |
> | Variance | 68.9 / 79.1 | 66.6 / 80.1 |
> | Energy | 98.1 / 39.9 | 97.7 / 44.2 |
>
>
> **ViT-B/16**
>
> | Method | Last Layer (FPR / AUROC) | Intermediate Fusion (FPR / AUROC) |
> |---|---|---|
> | MCM | 42.3 / 90.83 | 30.7 / 93.5 |
> | MaxLogit | 67.7 / 84.1 | 59.1 / 87.3 |
> | Entropy | 77.2 / 73.7 | 61.21 / 82.8 |
> | Variance | 76.9 / 74.1 | 60.9 / 83.0 |
> | Energy | 98.7 / 36.2 | 98.1 / 38.6 |
>
>
> ### **Q3: Validity of Random Projections for Geometry Preservation**
>
> As detailed in Appendix I, our use of fixed random projections is supported both theoretically and empirically. The Johnson–Lindenstrauss lemma [10, 32, 40] guarantees that such projections approximately preserve pairwise distances. Empirically, RanPAC [40] shows that random projections maintain the discriminative structure of visual features across domains, validating their use for aligning intermediate features without retraining.
>
> ### **Q4: Computational Overhead of Using Intermediate Layers**
>
> In response to the reviewer’s suggestion, we evaluated the computational overhead introduced by our method on both **ResNet** and **ViT-B/16**. At batch size 8 (averaged over 5 runs), **ResNet** shows a memory increase from **0.41 GB to 0.50 GB**, latency from **17.1 ms to 19.5 ms**, and a **12% drop in throughput** (58.6 → 51.4 img/s). For **ViT-B/16**, memory increases from **0.57 GB to 0.69 GB**, while latency remains stable (**57.5 ms to 54.5 ms**) and throughput is unchanged (**54.1 to 54.4 img/s**). These results support that, as requested, our method incurs **minimal additional cost**, particularly for transformer-based architectures. Full details will be included in the appendix.
>
>
> ### **Q5: Comparison to State-of-the-Art Methods**
>
> This is an interesting point, and we thank the reviewer for highlighting it. While recent methods such as **CSP**[4], **NegLabel** [5], and **CLIPScope** [6] achieve strong results, they typically rely on external resources such as WordNet hierarchies or handcrafted prompts, and use scoring functions specifically designed for their frameworks. In contrast, our method is orthogonal to these approaches. It focuses on the internal structure of the model and leverages intermediate-layer fusion, making it broadly compatible with other scoring strategies. To illustrate this, we applied our method to **NegLabel** and observed consistent improvements:
> | Method | FPR Avg ↓ | AUROC Avg ↑ | FPR SUN | AUROC SUN | FPR Places | AUROC Places | FPR DTD | AUROC DTD | FPR iNat | AUROC iNat |
> |---|---|---|---|---|---|---|---|---|---|---|
> | NegLabel | 25.40 | 94.21 | 20.53 | 95.49 | 35.59 | 91.64 | 43.56 | 90.22 | 1.91 | 99.49 |
> | **NegLabel + Int. Layers** | **23.79** | **95.05** | 21.10 | 95.38 | 32.31 | 93.35 | 39.38 | 92.10 | 2.35 | 99.36 |
>
> ### **Q6: Why Do Metrics Like Kurtosis, Gini, and JSD Perform Worse Than Random?**
>
> Heuristics like kurtosis, Gini, and JSD perform worse than random because they often select redundant or uninformative layers, typically shallow or final ones, which limits the diversity needed for effective OOD detection. In contrast, entropy consistently selects mid-to-deep layers (such as 6 to 8) that offer a better balance between abstraction and diversity, achieving lower FPRs (e.g., 0.307 compared to up to 0.435). This is supported by our SVCCA and agreement analyses (Figures 3 and 4), the entropy–FPR correlation in Figures G.11 and G.12, and the marginal utility study. Random selection, although unguided, sometimes includes strong layers by chance, which explains its surprisingly competitive average performance. For further details, we refer to **Response Q1 to Reviewer V3xf**, which compares selection strategies, and **Response Q2**, which analyzes the individual utility of each layer.

---

> > ### Comment · Reviewer_A4Qo · 2025-08-04
> >
> > Thank you for the response. However, I still have concerns regarding the experimental comparison with recent CLIP-based OOD detection methods. In particular, I would appreciate it if you could evaluate the proposed method against stronger baselines, such as CSP, to more clearly demonstrate its improvements.

---

> ### Author Response · Authors · 2025-08-04
> **Request for comparison against stronger CLIP-based methods such as CSP**
>
> Thank you for engaging. By combining CSP with intermediate layers, we observe a notable improvement, achieving lower FPR and higher AUROC:
>
> | Method                       | FPR Avg ↓ | AUROC Avg ↑ | FPR SUN | AUROC SUN | FPR Places | AUROC Places | FPR Textures | AUROC Textures | FPR iNat | AUROC iNat |
> | ---------------------------- | --------- | ----------- | ------- | --------- | ---------- | ------------ | ------------ | -------------- | -------- | ---------- |
> | CSP + Intermediate Layers | **15.52** | **96.74**   | 10.90   | 97.59     | 26.97      | 93.97        | 22.73        | 95.75          | 1.48     | 99.64      |
> | CSP                          | 17.51     | 95.76       | 13.66   | 96.66     | 29.32      | 92.90        | 25.52        | 93.86          | 1.54     | 99.60      |
>
> One point to highlight is that CSP already consumes ~17,654 MB of VRAM, and our method can be seamlessly integrated without introducing any additional memory or computational overhead. We will include these results in the final version of the paper.

---

> > ### Comment · Reviewer_A4Qo · 2025-08-05
> >
> > The authors’ response effectively addressed my concerns, and I will raise my score in the final evaluation.

---

### Official Review · Reviewer_rDhG · 2025-06-30

**Clarity:** 4
**Significance:** 2
**Originality:** 2
**Rating:** 5
**Confidence:** 4

**Summary:**

This paper presents an empirical exploration into the value of leveraging intermediate layers when using zero-shot OOD detection methods like MCM. In particular, the authors finds that intermediate layers can contain rich signals that can be leveraged for improving OOD detection beyond the currently standard final-layer OOD detection methods. The proposed updated MCM method which fuses intermediate layers is OOD agnostic and can be directly leveraged during inference. Additionally, the authors provide extensive empirical evaluations across multiple common pre-trained architectures.

**Questions:**

Additional questions beyond weaknesses listed above:
- The reviewer wonders if the authors have an intuitive understanding of why intermediate layers help particularly with near-OOD shifts.
- Have the authors attempted to see if the selection of intermediate layers can be further improved with OOD data?
- Any empirical evaluations on OOD detection tasks beyond image or on lower resolution CIFAR-based datasets?

**Ethical Concerns:**

["NO or VERY MINOR ethics concerns only"]

**Final Justification:**

This paper presents a thorough series of evaluations into the value of intermediate layers when performing OOD detection with MCM. The methodology is intuitive and readily applicable to real-world scenarios of OOD detections. Given the technic soundness and extensive evaluations, I will maintain my score to reflect my positive view on the paper's acceptance.

**Limitations:**

Yes

**Paper Formatting Concerns:**

The reviewer found no major formatting concerns.

**Quality:**

4

**Strengths And Weaknesses:**

The reviewer notes the following strengths and weaknesses.

Strengths:
- The paper presents a comprehensive empirical analysis on the value of intermediate layers when leveraging zero-shot OOD detection methods that is rarely explored.
- The proposed methodology is simply to use and can be directly applicable given the zero-shot nature of MCM and entropy-based selection of intermediate layers.
- The paper is well written and provides extensive empirical evaluations across multiple high resolution datasets and model architectures.

Weaknesses:
- It seems to the reviewer that OOD detection gains from intermediate layers is highly contingent on the underlying model, with CLIP-like models benefiting the most while others showing minimal benefits. This can limit the applicability of the methodology.
- The proposed methodology is an simple extension of MCM, which limits the novelty of the underlying proposed method.
- Much of the work, from the reviewer's perspective, is based on empirical evaluations and observations. Additional theoretical justifications may help improve the paper.

---

> ### Author Rebuttal · Authors · 2025-07-30
>
> We thank Reviewer rDhG for the thoughtful and encouraging feedback. Below, we address each of the reviewer’s questions in detail. We elaborate on the intuition behind the effectiveness of intermediate layers for near-OOD shifts, discuss potential extensions involving OOD supervision, and clarify our evaluation choices and directions for future work.
>
> ### **Q1: Why do intermediate layers help particularly with near-OOD shifts?**
>
> **Intermediate layers help in these cases because they capture diverse and complementary features that are often lost in the final layer**. As shown in Figure F.10, early layers focus on textures and materials, **middle layers on object parts and natural elements, while final layers emphasize high-level semantic concepts**. By fusing signals from intermediate layers, our method retains fine-grained information critical for distinguishing near-OOD samples, leading to improved performance on datasets such as NINCO, and SSB-Hard.
>
> [1] Oikarinen et al.: CLIP-Dissect: Automatic Description of Neuron Representations in Deep Vision Networks
>
> ### **Q2: Can intermediate layer selection be further improved using OOD data?**
>
> Thank you for the suggestion. While leveraging OOD data (e.g., Outlier Exposure \[2]) can enhance robustness, it alters the problem setting by requiring access to OOD samples. In contrast, to explore the potential of OOD usage without supervision, we conducted an oracle-based analysis using per-sample dynamic layer selection. The results show clear room for improvement: average FPR\@95 drops to **0.2004** for ViT-B/32 and **0.1680** for ViT-B/16. This highlights the potential of combining test-time adaptive strategies with OOD data as a promising future direction.
>
> [2] Hendrycks et al.: Deep Anomaly Detection with Outlier Exposure
>
>
> ### **Q3: Any empirical evaluations on OOD detection tasks beyond image or on lower resolution CIFAR-based datasets?**
>
> Yes. we also evaluate on Pascal VOC. Results are provided in Table 5.1. We do not use CIFAR-based benchmarks, as their low resolution images lack the semantic complexity needed for OOD detection with models like CLIP, and they are not widely used in recent CLIP-based OOD benchmarks.

---

> ### Comment · Reviewer_rDhG · 2025-08-05
> **Response**
>
> I would like to thank the authors for addressing my questions and clarifying the details. While I would still encourage the authors to consider strengthening their evaluation by including the CIFAR benchmark (given its common use in OOD detection) I do understand the reasoning behind its omission. Consequently, I will maintain my score as I continue to believe the work meets the standards for acceptance.

---

> > ### Author Response · Authors · 2025-08-08
> >
> > Thank you for taking the time to engage with our rebuttal and for your encouraging remarks. We truly appreciate your interest in the intuition behind near-OOD performance and your suggestions for broader evaluations.

---

### Official Review · Reviewer_Zjg4 · 2025-07-03

**Clarity:** 3
**Significance:** 3
**Originality:** 3
**Rating:** 5
**Confidence:** 4

**Summary:**

This paper presents a study on the impact of intermediate-layer representations in CLIP to OoD detection and show that certain intermediate layers can help improve the performance. Following, an entropy-based criterion is proposed to automatically select informative layers, and then fuse based on Maximum Concept Matching (MCM). Experimental setups are comprehensive.

**Questions:**

1. In line 233-234, the authors claim the performance of “24.63% / 93.93 on Pascal-VOC.” However, I can not find these performance figures in the table. Please let me know if I overlooked any details.
2. Can the entropy-based layer selection be extended to dynamically select layers per input sample, rather than per dataset?
3. How does this method generalize to other CNN-based or deeper vision encoder architectures?
4. What is the computation overhead of the proposed method?
5. How does this method perform with a real-world applications/dataset (e.g., medical imaging, autonomous driving)
6. Have you tried other statistical metrics for layer selection beyond entropy?

**Ethical Concerns:**

["NO or VERY MINOR ethics concerns only"]

**Final Justification:**

The rebuttal well addresses and clarifies my concerns.

Particularly, the additional results on Per-Instance Dynamic Layer Selection are interesting.

Therefore, I increase the final rating that this paper meets the acceptance standard.

**Quality:**

3

**Strengths And Weaknesses:**

Strengths:
* First study on the impact of intermediate-layer representations in CLIP to OoD detection
* The method is training-free, requiring no fine-tuning, labels, or OoD data.
* The evaluation is extensive including both far-OoD (ImageNet-1K, Pascal-VOC) and near-OoD (NINCO, SSB-Hard) benchmarks, multiple baselines such as MCM, GL-MCM, and SeTAR.
* The design is strongly motivated by interesting empirical findings

Weaknesses:
* The findings of the impact of intermediate-layer to OoD Detection is limited to CLIP or training-free OoD Detection Methods such as MCM
* While the improvement is significant compared to the baseline MCM, the advantage of the proposed method is minimal (or even lower performance) than SOTA training-free methods (See Tab. 1)
* Since this study related to model’s representations, evaluation on ViT-B16/32 might be limited. Will this method generalize to other CNN-based or deeper vision encoder architectures?
* I understand that the experimental setups are standard for training-free OoD detection. However, since the proposed method requires no training, it would be more interesting to evaluate on a critical real-world applications (e.g., medical imaging, autonomous driving) rather than ImageNet-1K or Pascal-VOC
* Minor: Technically, the proposed method requires multiple MCM inferences at different layers, the computation overhead is not discussed in the paper.

---

> ### Author Rebuttal · Authors · 2025-07-30
>
> We sincerely thank the reviewer for their thoughtful and encouraging feedback. We are pleased to hear that the writing, motivation, and experimental rigor of our work were appreciated. In response to the reviewer’s suggestions, including the evaluation on more architectures, per-instance selection, computational overhead, and comparisons to additional metrics, we conducted new experiments and analyses, which we detail below. Each point raised has been carefully addressed in the revised response.
>
>
> ### **Q1: Clarification on Pascal-VOC Results**
>
> Thank you for pointing this out. This was a typographical error in the text. The **correct performance values are those reported in Table 5.1**, which reflect the actual evaluation. We will update the manuscript to resolve this inconsistency in the final version.
>
>
> ### **Q2: Per-Instance Dynamic Layer Selection**
>
> We thank the reviewer for this valuable suggestion. While we did not implement per-instance dynamic selection in our current method, we conducted an **oracle-based analysis** to assess its potential. Specifically, for each architecture and dataset, we identified the **best-performing combination of layers per input sample**, allowing the selected layers to vary dynamically across instances.
>
> This yields the following **lower-bound FPR\@95** results:
>
> * **ViT-B/32:** iNat 0.0575, DTD 0.2472, SUN 0.2122, Places 0.2845 → **Avg FPR\@95: 0.2004**
> * **ViT-B/16:** iNat 0.0693, DTD 0.1397, SUN 0.2305, Places 0.2325 → **Avg FPR\@95: 0.1680**
>
> These results indicate that **per-sample dynamic layer selection could lead to further improvements**, particularly on challenging datasets. While our current method performs entropy-based selection at the dataset level, this oracle highlights a promising direction for future work on **test-time adaptive strategies**.
>
>
> ### **Q3: Generalization to CNNs and Deeper Vision Architectures**
>
> **Following the reviewer’s suggestion, we extended our evaluation to include both convolutional and deeper transformer-based vision architectures**. Specifically, we tested our method on **CLIP RN50x4** and **CLIP ViT-L/14**. The results below confirm that our approach generalizes well across different CLIP architectural types.
>
> **CLIP RN50x4 (ResNet)**
> | Method             | FPR Avg ↓ | AUROC Avg ↑ | FPR SUN | AUROC SUN | FPR Places | AUROC Places | FPR DTD | AUROC DTD | FPR iNat | AUROC iNat |
> | ------------------ | --------- | ----------- | ------- | --------- | ---------- | ------------ | ------- | --------- | -------- | ---------- |
> | MCM (Last Layer)   | 45.16     | 89.95       | 35.13   | 92.83     | 44.30      | 89.38        | 57.22   | 85.99     | 44.00    | 91.59      |
> | MCM (Intermediate) | **41.61** | **90.80**   | 31.83   | 93.48     | 40.46      | 90.36        | 52.38   | 87.37     | 41.79    | 92.00      |
>
> **CLIP ViT-L/14 (24-layer Transformer)**
> | Method             | FPR Avg ↓ | AUROC Avg ↑ | FPR SUN | AUROC SUN | FPR Places | AUROC Places | FPR DTD | AUROC DTD | FPR iNat | AUROC iNat |
> | ------------------ | --------- | ----------- | ------- | --------- | ---------- | ------------ | ------- | --------- | -------- | ---------- |
> | MCM (Last Layer)   | 37.16     | 91.66       | 29.58   | 93.98     | 35.51      | 92.02        | 58.65   | 85.19     | 24.91    | 95.44      |
> | MCM (Intermediate) | **37.09** | **91.70**   | 29.21   | 94.07     | 35.02      | 92.13        | 58.99   | 85.14     | 25.15    | 95.45      |
>
> #### **A: Model-Specific Behavior**
>
> While **ViT-L/14 shows smaller gains**, this aligns with our analysis in Section 3. Figures 2–4 and lines 142–146 show that ViT-L/14 exhibits **greater redundancy** and **flatter entropy profiles** across layers, limiting the benefits of fusion. This is further influenced by its **sharper softmax distribution**, which required a lower temperature setting (0.01). In contrast, **ResNet-based models like RN50x4 benefit more** from fusion due to **higher inter-layer diversity** and **smoother confidence dynamics**.
>
> ### **Q4: Computational Overhead**
>
> As requested, we performed additional experiments to quantify the computational overhead of our method. We evaluated inference time, memory usage, and throughput on ViT-B/32, comparing the use of the last layer versus our intermediate-layer fusion strategy. The results show that the use of intermediate layers introduces negligible overhead: memory usage increases by at most 0.04 GB, latency remains virtually unchanged, and throughput is preserved across all batch sizes.
>
> **ViT-B/32 | Last Layer vs Intermediate Layers**
> | Batch Size | Memory (GB) — Last | Memory (GB) — Ours | Latency (ms) — Last | Latency (ms) — Ours | Throughput (img/s) — Last | Throughput (img/s) — Ours |
> |------------|--------------------|--------------------|---------------------|---------------------|---------------------------|---------------------------|
> | 1          | 0.57               | 0.58               | 454.6 ± 25.8        | 452.4 ± 14.4        | 2.2 ± 0.1                 | 2.2 ± 0.1                 |
> | 2          | 0.57               | 0.58               | 227.5 ± 9.1         | 263.5 ± 79.8        | 4.4 ± 0.2                 | 4.1 ± 0.8                 |
> | 4          | 0.57               | 0.59               | 109.2 ± 3.7         | 108.0 ± 1.2         | 9.2 ± 0.3                 | 9.3 ± 0.1                 |
> | 8          | 0.58               | 0.62               | 54.5 ± 1.3          | 53.9 ± 1.1          | 18.4 ± 0.4                | 18.5 ± 0.4                |
>
>
> ### **Q5: How does the proposed method perform when applied to real-world scenarios or domain-specific?**
>
> Our method could be well suited for real-world applications where inference efficiency, scalability, and reliability are essential, based on the computational results shown in the previous table. Unlike methods that require retraining or computationally expensive ensembles, our approach is lightweight and don't need a retraining.
>
> Despite this, we acknowledge that real-world scenarios often involve challenges such as long-tailed class distributions and subtle distribution shifts factors known to significantly impact OOD detection performance [1, 2, 3]. While our current study focuses on standard benchmarks, the proposed method is modular, lightweight, and inference-only, making it a promising candidate for extension to safety-critical domains. We consider this an important direction for future work.
>
> **References**
> [1] Miao et al. *Out-of-Distribution Detection in Long-Tailed Recognition with Calibrated Outlier Class Learning*
> [2] Wei et al. *EAT: Towards Long-Tailed Out-of-Distribution Detection*
> [3] Miao et al. *Long-Tailed Out-of-Distribution Detection via Normalized Outlier Distribution Adaptation*
>
> ### **Q6: Have you tried other statistical metrics for layer selection beyond entropy?**
>
> Yes, we explored several metrics for layer selection, as shown in Section 6 and Figure 6. These include entropy, kurtosis, Gini, JSD, and random selection. **Entropy consistently performs best**, selecting mid-to-deep layers (e.g., 6, 8, 11) that offer complementary information and result in the lowest FPR and highest AUROC. In contrast, other metrics often favor shallow or final layers, leading to weaker performance. We will include additional metrics and visualizations in the revised version to provide more detailed comparisons.

---

> ### Comment · Reviewer_Zjg4 · 2025-08-06
> **Response to the rebuttal**
>
> I thank the authors for the rebuttal.
>
> For Q1, Q3, Q4, Q6, they are well addressed.
>
> For Q2, this is interesting and I encourage the authors to include and discuss more in the revision.
>
> For Q6, I understand that the time is limited for the rebuttal, but include some practical dataset setups would significantly strengthen the paper.
>
> Overall, I will increase the final rating.

---

### Official Review · Reviewer_WaeU · 2025-07-05

**Clarity:** 4
**Significance:** 3
**Originality:** 3
**Rating:** 5
**Confidence:** 3

**Summary:**

The authors propose a training-free strategy for OOD detection through using intermediate representations. They find that intermediate representations encode rich signals and propose entropy-based scoring for OOD detection which outperforms contemporary methods across various OOD detection benchmarks. The authors perform detailed experiments across ImageNet-1K and Pascal-VOC using CLIP with various backbones, and ablation studies over the hyperparameter and design choices of their method.

**Questions:**

How does your method perform across various architectures (& target modalities, such as perhaps text)?

I don't think this is an easy question to answer nor expect the authors to resolve it within the response period - but it is one that naturally comes up and would improve your work, if answered.

**Ethical Concerns:**

["NO or VERY MINOR ethics concerns only"]

**Final Justification:**

The paper is well written and tackles a relevant problem. My main perceived weakness was the lack of evaluation on text processing tasks, which would show that the method works across modalities. This weakness remains as it is (somewhat understandably) out of scope of the authors work. I consider the paper well written and rounded and worthy of publication.

**Limitations:**

Yes

**Quality:**

3

**Strengths And Weaknesses:**

Strengths
- The paper is very well written, with takeaways clearly highlighted and excellent visualizations.
- The experimental results are convincing
- The authors perform various ablation studies to analyse the robustness of their approach

Weaknesses
- The main weakness which if resolved, would take this paper a step beyond is evaluation on more models/architectures and possibly datasets

---

> ### Author Rebuttal · Authors · 2025-07-30
>
> We thank the reviewer for their thoughtful and encouraging feedback. We are glad the strengths of our work, including its clarity, experimental depth, and visualizations, resonated with you. In response to your main suggestion, we will expanded our evaluation and provide a more detailed discussion on performance across architectures and modalities. Below, we address your questions and comments point by point.
>
> ### **Q: Evaluation on more models/architectures**
>
> We do demonstrate that the method performs well **across a diverse range of vision architectures**. Our systematic evaluation covers 7 vision architectures of diverse types: supervised (ViT), self-supervised (MAE, MoCo-v3, DINOv2), and contrastive (CLIP, SiGLip-v2, Perception Encoder). This allows discovering consistent trends (see next answer). We propose to **include additional results from larger models**, namely CLIP ViT-L/14 and PEv2-L/14 in the final version of the paper.
>
>
> ### **Q: How does the method perform across architectures?**
>
> The systematic evaluation across architectures shows that contrastive models (e.g. CLIP) consistently benefit from intermediate-layer fusion due to their high inter-layer diversity and stable prediction profiles. In contrast, supervised and self-supervised models often exhibit redundant or unstable representations across layers, which limits the benefits of fusion. These differences are driven both by the training objective and the dataset scale, which together influence cross-layer complementarity.
>
> ### **Q: Other modalities**
>
> The extension to other modalities such as text would be very interesting because similar layer-wise computations are likely at play. Indeed, prior work like Layer-by-Layer [1] and DoLa [2] shows that intermediate representations can improve performance and reduce hallucinations in LLMs. Some modality-specific adaptations would be required, for example replacing patch-based fusion with token-level strategies. We propose to add relevant discussion and suggestions in the "Future Work" section.
>
> [1] Skean et al. *Layer by Layer: Uncovering Hidden Representations in Language Models*
>
> [2] Chuang et al. *DoLa: Decoding by Contrasting Layers Improves Factuality in Large Language Models*

---

> ### Author Response · Authors · 2025-08-08
>
> We're glad to hear that the clarity, visualizations, and experimental depth of our work resonated with you. We appreciate your suggestions regarding evaluation across architectures and modalities, and we’re happy our responses addressed your concerns. Thank you for your engagement and for maintaining your positive evaluation.

---

### Note · Authors · 2025-08-13

We thank all reviewers for their constructive feedback and for recognizing the clarity, breadth, and rigor of our study. Revisions and new experiments during the rebuttal addressed all concerns, and we are especially grateful for the requests for **additional analyses and comparisons**, which directly strengthened the paper.

**Key Contributions:**

* First **systematic, training-free** study showing that **intermediate-layer representations provide complementary information** to the final layer, improving zero-shot OOD detection across **seven diverse vision architectures** (supervised, self-supervised, and contrastive).
* **Interoperable fusion**: Intermediate-layer fusion **enhances strong existing methods** (e.g., NegLabel, CSP) and **works with multiple scoring rules** (MCM, MaxLogit, entropy, variance, energy), indicating that gains stem from representation complementarity rather than any specific score.
* **Architecture-aware insights** explain when and why gains emerge, supported by agreement maps, SVCCA, and marginal-utility analyses that highlight the value of mid–deep layers.
* **Practical and efficient**: The approach remains inference-only, training-free, and introduces **negligible overhead** in memory and latency.

**Summary of Rebuttal Additions:**

* **Broader architectures**: Added CLIP RN50x4 and ViT-L/14, confirming generalization to convolutional and deeper transformer models.
* **Stronger baselines**: Integrated with **CSP and NegLabel**, yielding **further FPR reductions and AUROC gains** without added runtime cost.
* **Broader scoring**: Verified that complementarity holds across diverse scoring strategies.
* **Overhead benchmarks**: Detailed profiling of memory, latency, and throughput showing minimal cost.
* **Layer behavior**: Clarified selection patterns and why complementary mid–deep layers drive the improvements.
* **Per-instance oracle**: Demonstrated additional headroom for **test-time adaptive fusion**.

We thank the reviewers again for their thoughtful dialogue. Your suggestions led to new experiments, richer analyses, and a stronger, more generalizable final version of the paper. Thank you!

---

### Decision · Program_Chairs · 2025-09-17

**Decision:**

Accept (poster)

**Comment:**

This paper investigates the role of intermediate-layer representations in pre-trained vision and vision-language models for out-of-distribution (OOD) detection. The authors propose an entropy-based layer selection method and show that fusing intermediate layers can improve zero-shot OOD detection performance across diverse architectures. The work is training-free, inference-only, and systematically studied across 7 backbones. The rebuttal period added new experiments, including integration with stronger baselines (CSP, NegLabel), evaluation on larger and convolutional architectures, per-instance oracle analysis, overhead profiling, and expanded comparisons across scoring strategies.

Overall, the reviews converge on the paper being technically sound, clearly written, and empirically comprehensive, with several reviewers explicitly raising their scores after the rebuttal. Concerns about limited novelty are valid, but the rebuttal effectively demonstrated that the method is orthogonal and complementary to existing state-of-the-art approaches. The authors are encouraged to take reviewers' suggestions to further improve the manuscript.